# Late Miocene climate cooling and intensification of southeast Asian winter monsoon

Ann E. Holbourn [1], Wolfgang Kuhnt [1], Steven C. Clemens [2], Karlos G.D. Kochhann [1,3], Janika Jöhnck [1], Julia Lübbers [1] & Nils Andersen [4]

The late Miocene offers the opportunity to assess the sensitivity of the Earth's climate to orbital forcing and to changing boundary conditions, such as ice volume and greenhouse gas concentrations, on a warmer-than-modern Earth. Here we investigate the relationships between low- and high-latitude climate variability in an extended succession from the subtropical northwestern Pacific Ocean. Our high-resolution benthic isotope record in combination with paired mixed layer isotope and Mg/Ca-derived temperature data reveal that a long-term cooling trend was synchronous with intensification of the Asian winter monsoon and strengthening of the biological pump from ~7 Ma until ~5.5 Ma. The climate shift occurred at the end of a global $\delta^{13}C$ decrease, suggesting that changes in the carbon cycle involving the terrestrial and deep ocean carbon reservoirs were instrumental in driving late Miocene climate cooling. The inception of cooler climate conditions culminated with ephemeral Northern Hemisphere glaciations between 6.0 and 5.5 Ma.

[1] Institute of Geosciences, Christian-Albrechts-University, Kiel D-24118, Germany. [2] Earth, Environmental and Planetary Sciences, Brown University, Box 1846, Providence, RI 02912, USA. [3] Technological Institute of Micropaleontology, Unisinos University, São Leopoldo 93022-750, Brazil. [4] Leibniz Laboratory for Radiometric Dating and Stable Isotope Research, Christian-Albrechts-University, Kiel D-24118, Germany. Correspondence and requests for materials should be addressed to A.E.H. (email: ah@gpi.uni-kiel.de)

The late Miocene (~11.6 to 5.3 Ma) stands out as a period of exceptional interest within the long-term Cenozoic cooling trend toward icehouse conditions, as it represents a geologically recent interval of relative global warmth that was marked by profound environmental change in both terrestrial and marine ecosystems (e.g., refs. [1,2]). This interval provides a unique opportunity to document climate-carbon cycle dynamics on a warmer-than-modern Earth and, thus, to help guide models and constrain predictions of climate change and sensitivity. The detailed sequence of climate events and the range of natural climate variability through the late Miocene remain, however, poorly understood, mainly due to the scarcity of continuous, high-resolution climate archives. Most available records are adequate for characterizing long-term trends or mean states, but do not capture short-term climate events and orbital-scale phase relationships required to assess, for example, changes in ice volume, monsoon intensity, and carbon fluxes.

Multiproxy temperature reconstructions indicated that a reduced sea surface temperature (SST) zonal gradient generally prevailed in the tropical Pacific Ocean during the late Miocene, in contrast to the sharper gradient that developed during the late Pliocene to Pleistocene[3,4]. This mean state, typically referred to as "permanent El Niño-like conditions" or "El Padre"[5,6], exerts a fundamental impact on regional and global climate because it is dynamically linked to the weakening of the Hadley and Walker circulation and the state of upper-ocean stratification[4,7,8]. However, it is difficult to reconcile the late Miocene warmth with inferred low atmospheric $pCO_2$ levels, close to preindustrial values (e.g., ref. [9]). This apparent decoupling between climate warmth and atmospheric $pCO_2$ variations has prompted intense debate about the dynamics of warm climates and the role of $pCO_2$ as driver of climate variations under different background states (e.g., refs. [10,11]). The widely held view of sustained equable warmth through the late Miocene was recently challenged by SST reconstructions, which revealed that a prolonged global cooling spell occurred between ~7 and ~5.5 Ma[2,12]. During this period, SST dipped below early Pliocene values in the Mediterranean, the Pacific, Atlantic and Indian Oceans and the latitudinal thermal gradient intensified[2,12]. Land records also indicate substantial aridification and vegetation changes between ~8 and 5.5 Ma

(see compilation in ref. [13] and references therein) with a reversal of this long-term trend 5.3 Myr ago, at the beginning of the Pliocene period[14]. However, the drivers of these major changes are still enigmatic. It remains unclear, in particular, whether climate cooling occurred as a response to changing climate boundary conditions, such as ice volume, $pCO_2$, tectonic setting and ocean–atmosphere circulation and to what extent these changes were coupled to Northern or Southern Hemisphere climate dynamics.

In this study, we reconstruct the detailed evolution of deep and near surface water masses (using stable isotopes and mixed layer temperatures) at Ocean Drilling Program (ODP) Site 1146 in the South China Sea (Fig. 1) and we investigate relationships to high-latitude climate variability, global ocean circulation changes, and radiative forcing over the interval 9–5 Ma. This work extends published high-resolution benthic isotope and upper-ocean temperature time series from the same site[15–18]. During the late Miocene, the location and water depth of Site 1146 were approximately similar to that of today (19°27.40′ N, 116°16.37′ E, water depth: 2092 m) and the connection between the South China Sea and the western Pacific Ocean remained fully open[19,20]. The benthic isotope signal at Site 1146 is, therefore, representative of Pacific intermediate/deep water masses originating at higher latitudes. Site 1146 is located at the northwestern edge of the western Pacific warm pool (WPWP) and is also ideal to constrain meridional variations in the extent of the WPWP and to monitor changes in southeast Asian monsoon climate. The extended, carbonate and clay-rich succession recovered at this site[20], thus, provides an outstanding archive of subtropical climate variations, allowing new insights into the dynamics and forcing processes of late Miocene climate evolution.

## Results

**Late Miocene astronomically tuned chronology.** The 1146 benthic foraminiferal stable isotope records based on *Cibicidoides wuellerstorfi* and/or *Cibicidoides mundulus* (5 cm sample spacing along a composite sequence or splice from Holes 1146A and 1146C) were tuned to an eccentricity-tilt (ET) composite target generated from the La2004 orbital solution[21] (Supplementary Note 1; Supplementary Figs. 1–4). The tuned series exhibits a

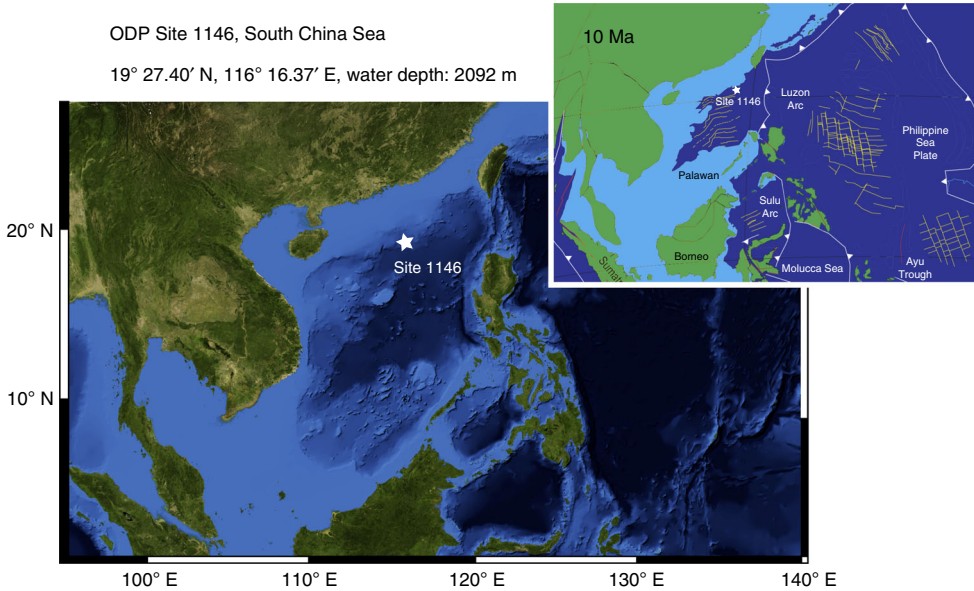

**Fig. 1** Location of ODP Site 1146 within a slope basin at the northern margin of the South China Sea. Satellite image and bathymetry from ref. [75]. Paleogeographic reconstruction at 10 Ma (simplified from ref. [19])

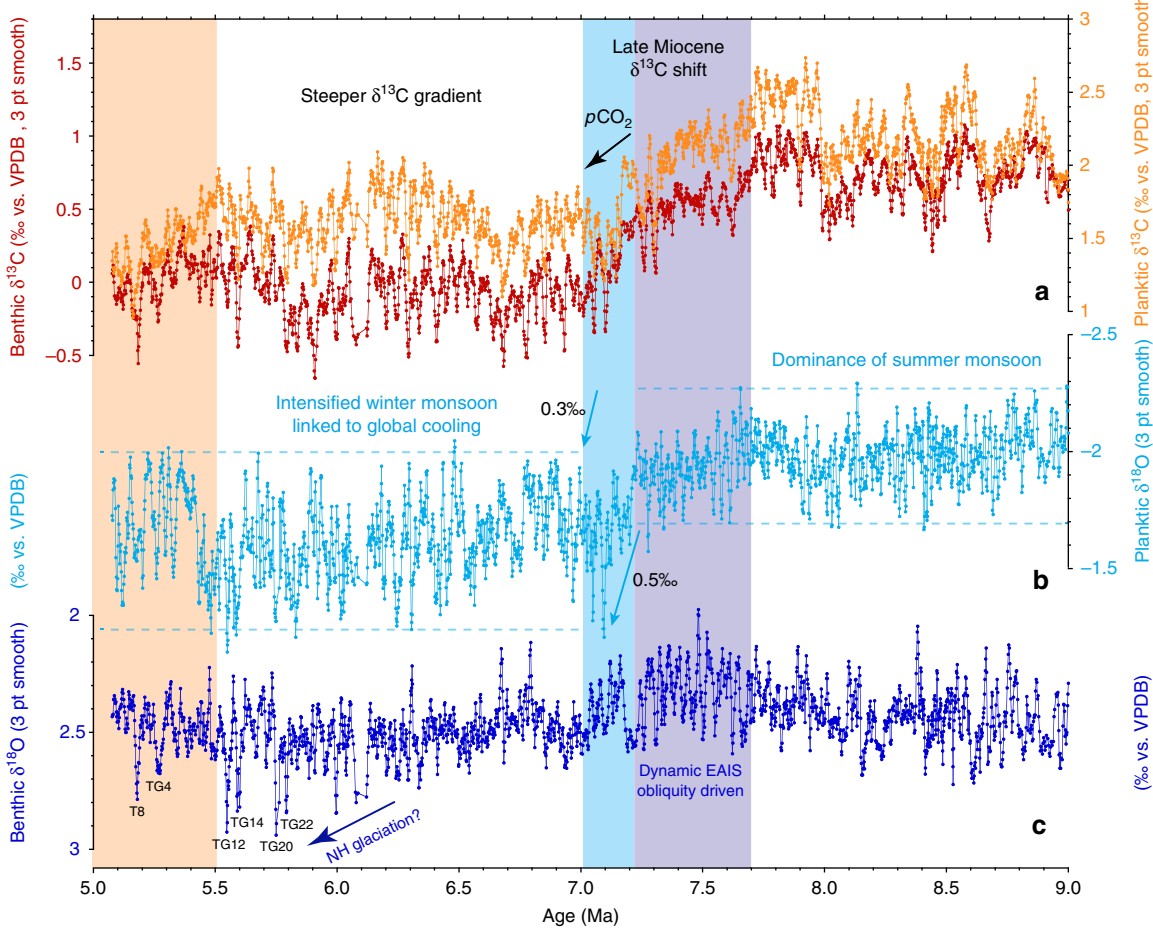

**Fig. 2** Late Miocene paleoceanographic records from ODP Site 1146. **a** Planktic (*G. sacculifer*) and benthic (*C. wuellerstorfi* and/or *C. mundulus*) $\delta^{13}$C. **b** Planktic (*G. sacculifer*) $\delta^{18}$O; dashed lines indicate amplitude variability over intervals 9–7.23 Ma and 7.03–5.07 Ma. **c** Benthic (*C. wuellerstorfi* and/or *C. mundulus*) $\delta^{18}$O. Lilac shading marks global $\delta^{13}$C decline coincident with planktic $\delta^{18}$O increase and high-amplitude obliquity modulation of benthic $\delta^{18}$O. Blue shading marks final stage of global $\delta^{13}$C decline. Light orange shading marks climate warming after 5.5 Ma. EAIS: East Antarctic ice sheet; 3 pt smooth: 3 pt moving mean

mean sedimentation rate of ~3 cm kyr$^{-1}$ with a maximum of 5 cm kyr$^{-1}$ and a minimum of 1 cm kyr$^{-1}$ and a mean temporal resolution of ~2 kyr over the interval 9–5 Ma (Supplementary Fig. 4B). We note that the short eccentricity (100 kyr) period is prominent in the untuned and tuned benthic $\delta^{18}$O records from 9.0 to 7.9 Ma and that the low amplitude of short eccentricity and high amplitude of obliquity between ~7.7 and 7.2 Ma are clearly reflected in the benthic $\delta^{18}$O series, which exhibits pronounced 41 kyr variability over this interval (Fig. 2c; Supplementary Figs. 1–4). The interval 6–5 Ma includes prominent benthic $\delta^{18}$O maxima, identified as T8, TG4, TG12, TG14, TG20, and TG22. These globally traceable $\delta^{18}$O enrichments[22–26] provide additional stratigraphic control. Superimposed on higher frequency variations (mainly 41 kyr), the untuned benthic and planktic $\delta^{13}$C series display low-frequency oscillations that broadly relate to the ~400 kyr long eccentricity cycle (Fig. 2a; Supplementary Fig. 1). Comparison of benthic and planktic $\delta^{18}$O and $\delta^{13}$C records plotted in the depth and time domains shows that the original spectral characteristics are preserved following the tuning procedure.

**Temporal trends in benthic and planktic stable isotopes.** Benthic and planktic $\delta^{18}$O exhibit different long-term trends

and short-term variability from ~9 to ~5 Ma (Fig. 2b, c; Supplementary Fig. 2A, B). Between 9.0 and 7.3 Ma, mean benthic $\delta^{18}$O varies between 2.5 and 2.3‰ and displays an overall decreasing trend of ~0.2‰ (Fig. 2c, Supplementary Fig. 4B) with standard deviations (SDs) ranging between 0.20 and 0.08‰ (Supplementary Fig. 2E). Lowest mean benthic $\delta^{18}$O values of 2.3‰ are reached between 7.7 and 7.3 Ma during an interval of pronounced 41 kyr variability (SD mainly between 0.16 and 0.12‰) (Supplementary Fig. 2B, E). In contrast, planktic $\delta^{18}$O oscillates around a mean of −2.0‰ from 9.0 to 7.7 Ma and exhibits a slight increasing trend to −1.9‰ from 7.7 to 7.3 Ma (Fig. 2b; Supplementary Fig. 2A). From 9.0 to 7.3 Ma, SDs in planktic $\delta^{18}$O fluctuate between 0.09 and 0.19‰ around a mean of 0.13‰ (Supplementary Fig. 2C).

Between 7.3 and 6.9 Ma, the Site 1146 high-resolution benthic and planktic $\delta^{18}$O records reveal a series of previously unrecognized short-term climate events (Fig. 2b, c). The benthic $\delta^{18}$O curve resolves a ~80 kyr long positive excursion (~0.3‰ amplitude) centered at 7.2 Ma followed by a rebound before a stepwise increase at 7.1–7.0 Ma (0.2‰ mean increase) (Fig. 2c; Supplementary Fig. 2B). The benthic $\delta^{18}$O shift was coupled to a stepwise increase in planktic $\delta^{18}$O between 7.2 and 7.1 Ma (Fig. 2b, c; Supplementary Fig. 2A, B), which marked the onset of a long-term trend of substantially heavier values (0.3‰ mean

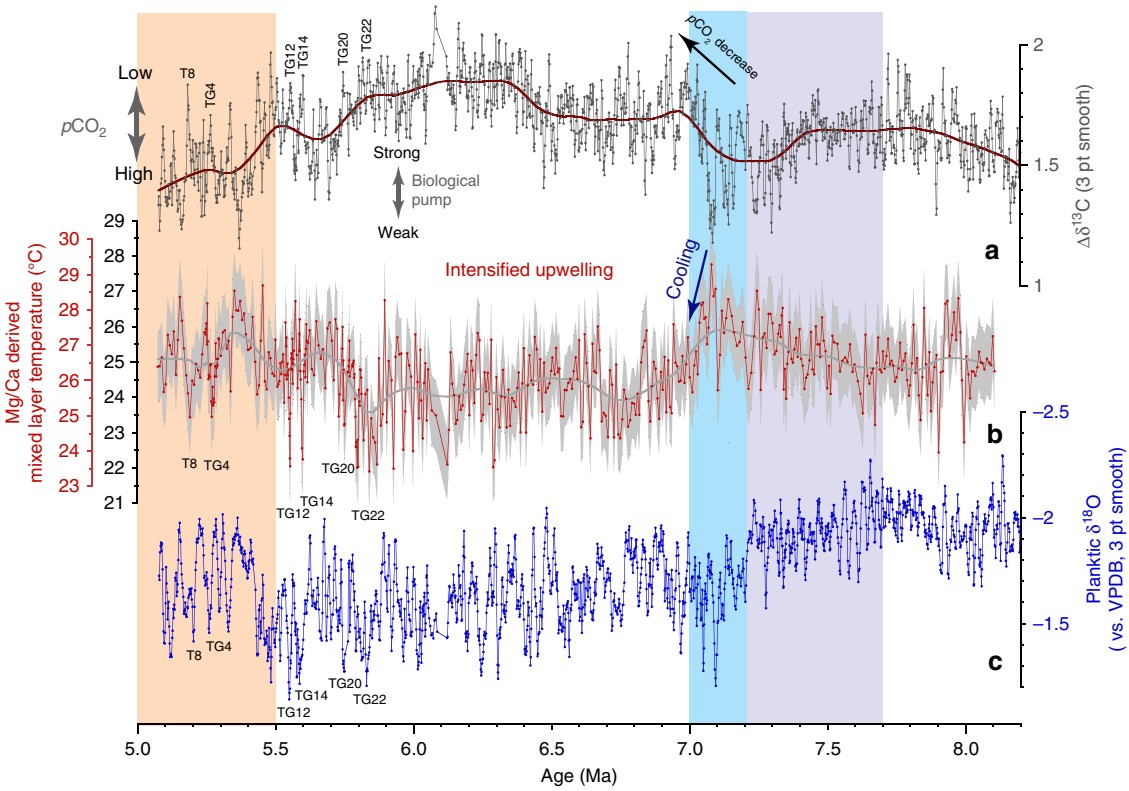

**Fig. 3** Evolution of mixed layer hydrology and productivity at ODP Site 1146. Steepening of $\Delta\delta^{13}C$ in conjunction with cooler temperatures and heavier planktic $\delta^{18}O$ between ~7 and 5.5 Ma denote $pCO_2$ decrease associated with enhanced carbon sequestration flux and intensification of Asian winter monsoon. **a** Gradient between planktic and benthic foraminiferal $\delta^{13}C$. **b** Mixed layer Mg/Ca-derived temperatures with (red scale) and without (black scale) correction for secular changes in Mg/Ca of seawater. Calculation of error envelope follows ref. [72]. Brown smooth curve in **a** and gray smooth curve in **b** fitted using locally weighted least squared error (Lowess) method. **c** Planktic foraminiferal $\delta^{18}O$; 3 pt smooth: 3 pt moving mean. Lilac shading marks global $\delta^{13}C$ decline coincident with planktic $\delta^{18}O$ increase and high-amplitude obliquity modulation of benthic $\delta^{18}O$. Blue shading marks final stage of global $\delta^{13}C$ decline. Light orange shading marks climate warming after 5.5 Ma

increase) and overall higher amplitude variability (mean SD 0.18‰ after 7.0 Ma vs. 0.13‰ prior to 7.2 Ma) (Supplementary Fig. 2A, C).

From ~7 to 5.2 Ma, mean benthic $\delta^{18}O$ oscillates around 2.5‰ (Supplementary Fig. 2B). Amplitude variability is relatively low until 6.1 Ma except for a few transient minima (mean SD 0.11‰ fluctuating between 0.07 and 0.16‰), but it increases markedly from 6.1 to 5.5 Ma (mean SD 0.14‰,), culminating in the high-amplitude maxima TG12, TG14, TG20, and TG22, when peak benthic $\delta^{18}O$ values reach ~3‰ (Fig. 2c; Supplementary Fig. 2E, F). Mean planktic $\delta^{18}O$ shows a slight increasing trend (~0.2‰ mean increase to a maximum of −1.5‰) after ~7 Ma, followed by a decrease to mean values between −1.8 and −1.6‰ after 5.5 Ma (Supplementary Fig. 2A). In contrast to benthic $\delta^{18}O$, the amplitude variability of planktic $\delta^{18}O$ (Supplementary Fig. 2C, D) increases markedly after ~7 Ma and generally remains high until 5.2 Ma (mean SD 0.18‰, fluctuating between 0.10 and 0.24‰).

Benthic and planktic $\delta^{13}C$ exhibit consistent long-term (400 kyr) and short-term (41 kyr) variability from ~9 to ~5 Ma (Fig. 2a; Supplementary Figs. 5, 6A). Between 9 and 7.7 Ma, benthic and planktic $\delta^{13}C$ oscillate between 1.1 and 0.1‰ (mean 0.73‰, SD 0.19‰) and between 2.8 and 1.6‰ (mean 2.21‰, SD 0.24‰), respectively. Between ~7.7 and ~7.0 Ma, a characteristic feature of the benthic and planktic $\delta^{13}C$ records is the massive, long-term decrease of >1‰, from 1.0 to −0.3‰ (benthic) and 2.7 to 1.3‰ (planktic), which corresponds to the global decline in $\delta^{13}C$ known as the late Miocene carbon isotope shift[27,28] (LMCIS, Fig. 2a). The final phase of the LMCIS at 7.2–7.0 Ma coincides

with a distinct sharpening of the gradient between planktic and benthic $\delta^{13}C$ ($\Delta\delta^{13}C$) between 7.1 and 7.0 Ma, which lasts until ~5.5 Ma (Figs. 3a and 4b), and with a stepwise increases in benthic and planktic $\delta^{18}O$ (Figs. 2b, c, and 4e, f). Between 7 and 5 Ma, following the end of the LMCIS, benthic and planktic $\delta^{13}C$ fluctuate between 0.47 and −0.74‰ (mean −0.05‰, SD 0.20‰) and between 2.16 and 0.88‰ (mean 1.59‰, SD 0.21‰), respectively (Fig. 2a).

Coherence ($k$) between benthic and planktic $\delta^{18}O$ remains overall lower than between benthic and planktic $\delta^{13}C$ over the interval 9–5 Ma, with a maximum of 0.90 on the obliquity band at a frequency of 0.02455 cycles kyr$^{-1}$ and a second maximum on the precession band of 0.83 at a frequency of 0.0425, whereas coherence on the 400 and 100 kyr eccentricity bands remains insignificant at the 80% level ($k$ below 0.4) (Supplementary Note 2; Supplementary Figs. 5, 6A, B). In contrast, benthic and planktic $\delta^{13}C$ exhibit high coherence both in their long-term (400 kyr) and short-term (predominantly 41 kyr) variability throughout the interval 9–5 Ma (Supplementary Note 2; Supplementary Fig. 6A, B), implying that both are influenced by changes in the global carbon cycle. Coherence on the long eccentricity band (frequencies of 0.0023–0.0029 cycles kyr$^{-1}$) fluctuates around 0.9, while coherence on the short eccentricity (0.0097 cycles kyr$^{-1}$) and obliquity (0.247 cycles kyr$^{-1}$) bands reaches maxima of 0.98 and 0.96, respectively.

**Evolution of mixed layer temperatures.** Reconstructed mixed layer temperatures based on Mg/Ca ratios in the planktic

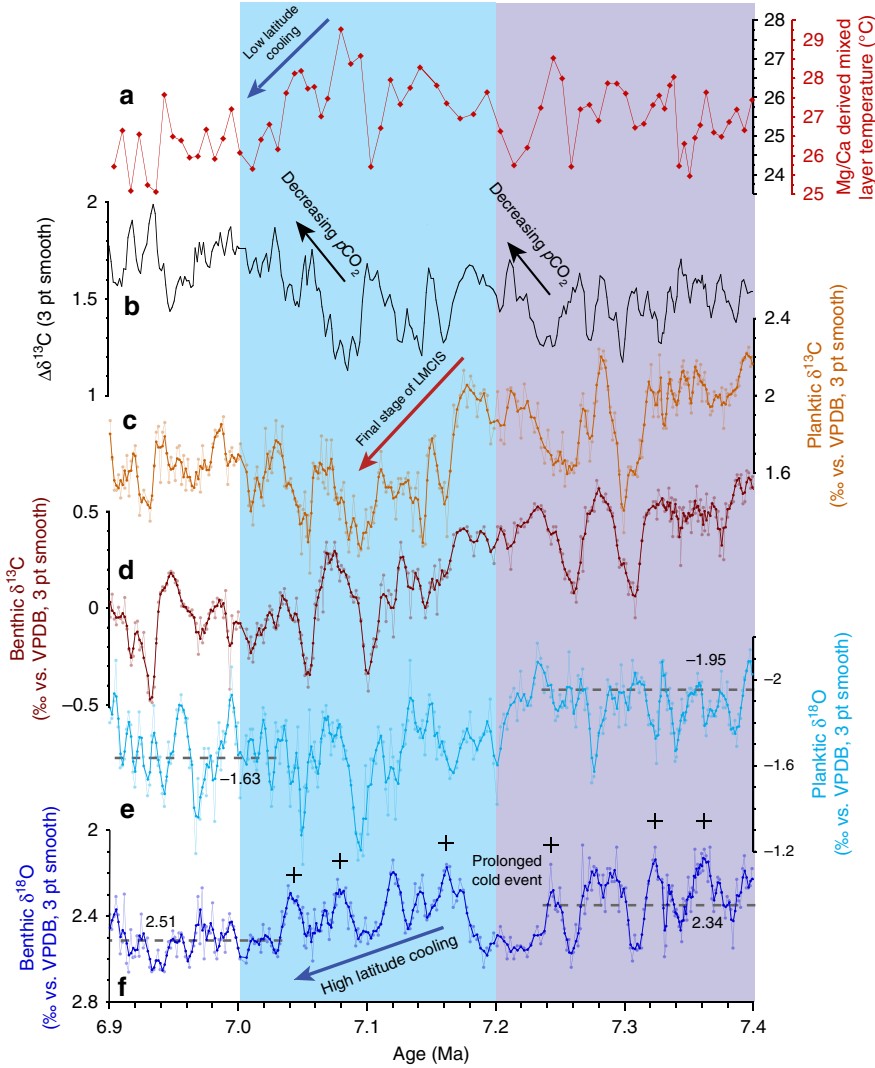

**Fig. 4** Expanded view of interval 7.4–6.9 Ma at ODP Site 1146. **a** Mg/Ca-derived mixed layer temperatures with (red scale) and without (black scale) correction for secular changes in Mg/Ca of seawater. **b** $\Delta\delta^{13}$C; **c** Planktic $\delta^{13}$C. **d** Benthic $\delta^{13}$C. **e** Planktic $\delta^{18}$O. **f** Benthic $\delta^{18}$O. Dashed horizontal lines indicate long-term mean of planktic and benthic foraminiferal $\delta^{18}$O for intervals 5.07–7.03 Ma ($n = 1114$ and 1015) and 7.23–8.00 Ma ($n = 454$ and 446). Lilac shading marks global $\delta^{13}$C decline coincident with planktic $\delta^{18}$O increase and high-amplitude obliquity modulation of benthic $\delta^{18}$O. Blue shading marks final stage of global $\delta^{13}$C decline. Age correlation points are indicated by crosses; 3 pt smooth: 3 pt moving mean

foraminifer *Globigerinoides sacculifer* (without correction for secular changes in seawater Mg/Ca concentrations) vary between 22 and 28 °C from 8.2 to 5 Ma (Fig. 3b). These relatively low values are most likely due to long-term secular changes in seawater Mg/Ca concentrations (e.g., ref. [29]). A correction for seawater Mg/Ca concentration, using a latest Miocene seawater Mg/Ca ratio of ~4.5 mol mol$^{-1}$ [29] and a modern day seawater Mg/Ca of ~5.1 mol mol$^{-1}$ [30] following the calculation outlined in ref. [31] results in a temperature increase of ~1.5 °C. This is consistent with estimates of a 4.3 mol mol$^{-1}$ Pliocene Mg/Ca ratio and 0.9–1.9 °C Pliocene Mg/Ca ocean temperatures, deducted from coupled seawater-test Mg/Ca temperature laboratory calibrations of *Globigerinoides ruber* (ref. [32]). Corrected temperatures (Figs. 3b, 4a, and 5a) are close to the range of modern seasonal variability in the area of Site 1146, between 24.7 and 27.8 °C at 25 m water depth[33] (Supplementary Note 3; Supplementary Fig. 7) with the warmest reconstructed temperatures exceeding modern mixed layer temperatures by almost ~2 °C, whereas the lowest temperatures during transient cold events remain ~1 °C below modern winter temperatures.

Mixed layer temperature estimates at Site 1146 exhibit a slight warming trend between 7.3 and 7.1 Ma, increasing from (uncorrected) mean values of 25.0 °C ($n = 122$, SD 0.7 °C) between 8.1 and 7.3 Ma to 25.8 °C ($n = 36$; SD 0.9 °C) between 7.3 and 7.1 Ma (Fig. 3b). This transient warming is interrupted by a pronounced cooling step of ~2 °C between 7.1 and 6.9 Ma, previously documented at this site by a low-resolution study[34]. This cooling step marks the onset of a long-term trend of cooler temperatures, lasting until ~5.5 Ma, which coincides with a long-term increase in the mean and amplitude variability of seawater $\delta^{18}$O (Supplementary Figs. 8, 9A, C) and with an intensification of the gradient between planktic and benthic $\delta^{13}$C (Figs. 3a and 4b). Between 6.9 and 5.7 Ma, uncorrected temperatures fluctuate between 21.9 and 26.8 °C with a mean of 24.2 °C ($n = 199$, 1.0 °C SD). After ~5.7 Ma, uncorrected temperatures generally oscillate between 22.1 and 27.0 °C and exhibit a sustained increase of 0.8 °C in their mean value ($n = 146$, 0.9° SD). Between 6.0 and 5.5 Ma, transient episodes of intense mixed layer cooling by ~2 to ~3 °C coincide with sharp increases in planktic $\delta^{18}$O and the prominent benthic $\delta^{18}$O maxima TG12, TG14, TG20, and TG22 (Fig. 5a, e, f).

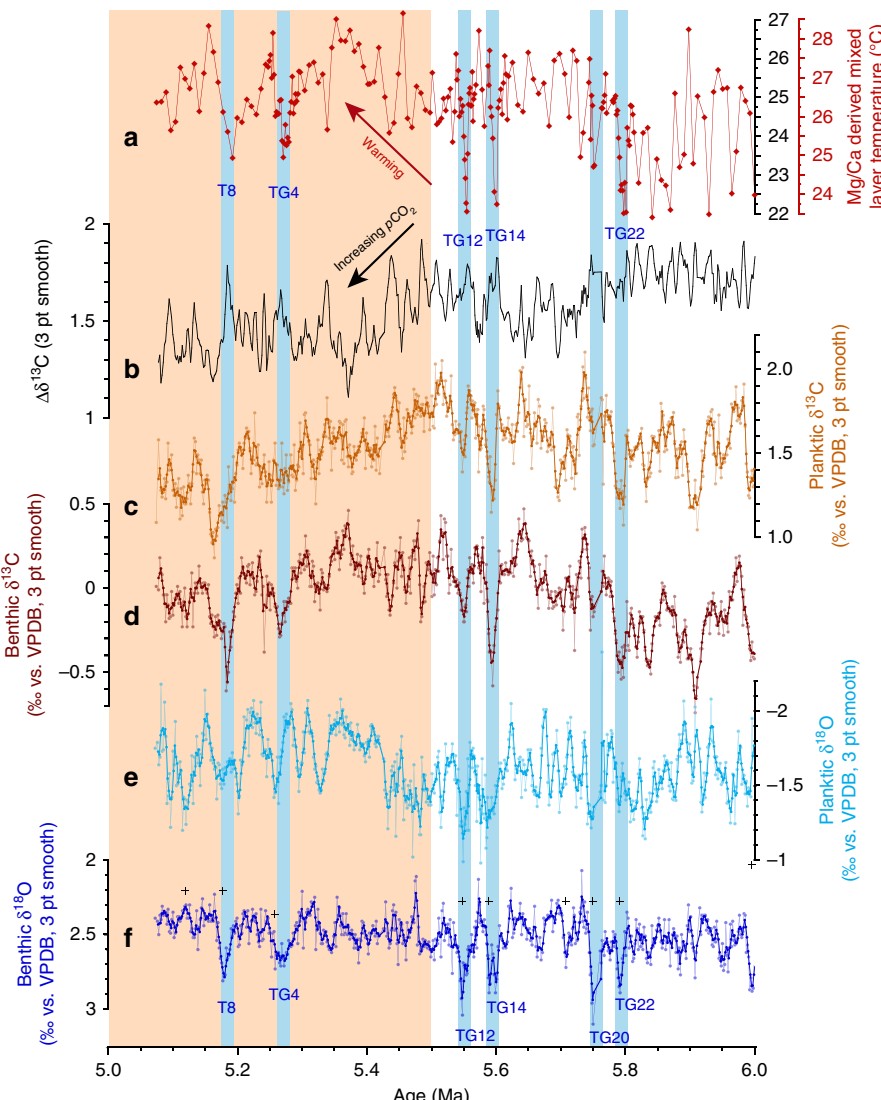

**Fig. 5** Expanded view of interval 6–5 Ma at ODP Site 1146. **a** Mg/Ca-derived mixed layer temperatures with (red scale) and without (black scale) correction for secular changes in Mg/Ca of seawater. **b** $\Delta\delta^{13}C$; **c** Planktic $\delta^{13}C$. **d** Benthic $\delta^{13}C$. **e** Planktic $\delta^{18}O$. **f** Benthic $\delta^{18}O$. Blue shading indicates intense transient cooling episodes associated with planktic $\delta^{13}C$ depletion and with planktic and benthic $\delta^{18}O$ enrichments (TG events). Light orange shading marks climate warming after 5.5 Ma. Age correlation points are indicated by crosses; 3 pt smooth: 3 pt moving mean

In this study, we discuss relative changes in mixed layer temperatures rather than absolute values, since the history of Miocene seawater Mg/Ca composition is still poorly constrained (Supplementary Note 3). Furthermore, our interpretations are based on relatively short-term changes in mixed layer temperatures, which are not affected by the long-term variability of seawater Mg/Ca concentration with changes in the order of millions of years due to the long residence time of these elements in the ocean[29].

## Discussion

The planktic and benthic $\delta^{18}O$ signals at Site 1146 differ markedly in their long- and short-term trends between 9 and 5 Ma, pointing to a decoupling of regional hydrology and the evolution of the Antarctic ice sheet, which formed the main component of the cryosphere during the middle and late Miocene (e.g., refs. [35,36]). Mixed layer temperature and seawater $\delta^{18}O$ reconstructions at Site 1146 additionally support that substantial changes in southeast Asian hydroclimate occurred after ~8 Ma, which accelerated at ~7 Ma, but do not appear closely connected with Southern Hemisphere high-latitude climate (benthic $\delta^{18}O$) trends.

Between 7.1 and 6.9 Ma, upper-ocean temperatures at Site 1146 document a sustained cooling (~2 °C mean cooling), which persisted until ~5.7 Ma (Figs. 3b and 4a). This cooling was associated with a long-term increase in the mean and amplitude variability of seawater $\delta^{18}O$ (Supplementary Note 3; Supplementary Fig. 9A, C), as indicated by a previous low-resolution study[34]. These trends signal a change in the amount and/or $\delta^{18}O$ composition of precipitation and runoff, likely associated with changes in the provenance and/or seasonality of precipitation toward a more pronounced monsoonal seasonality and a more temperature-controlled seasonality of rainwater $\delta^{18}O$ (i.e., $\delta^{18}O$ depleted winter precipitation[37]). We attribute these hydrological changes in the northern South China Sea after ~7 Ma to cooling and drying of the Asian landmass and a related southward shift of the average summer position of the Intertropical Convergence Zone (ITCZ), resulting in decreased influence of tropical convection and intensified dry winter monsoon over southeast Asia. Drying and cooling on the Asian continent at ~7 Ma are supported by independent lines of evidence including enhanced dust accumulation rates in northern China[38], vegetation change in central China[39] and an increase in the mean grain size of the terrigenous

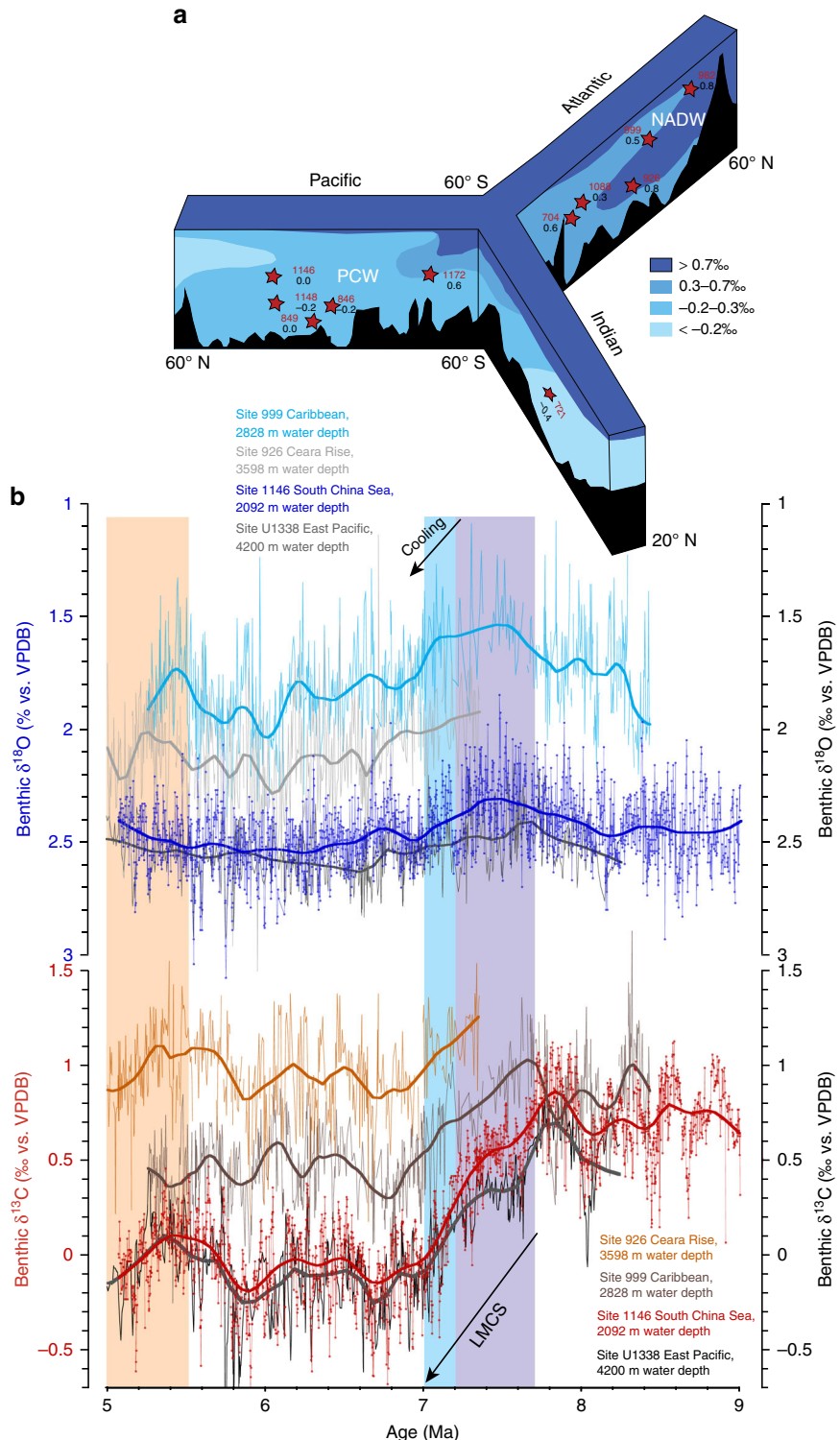

**Fig. 6** Comparison of late Miocene inter-basinal benthic $\delta^{18}O$ and $\delta^{13}C$ gradients. **a** Vertical distribution of $\delta^{13}C$ in world's oceans following late Miocene $\delta^{13}C$ shift. Averaged values over interval 7–5 Ma for key sites in the Pacific, Atlantic, Indian, and Southern Oceans compiled from refs. [22,54,55,76,] NADW: North Atlantic Deep Water, PCW: Pacific Central Water. **b** Late Miocene evolution of inter-basinal benthic $\delta^{18}O$ and $\delta^{13}C$ gradients: comparison of Pacific ODP Site 1146 and Atlantic ODP Sites 926 and 999[22,23] and equatorial Pacific IODP Site U1338[55] over interval 9–5 Ma. Stable isotope data from Sites 926 and 999 are plotted on originally published age models. Over the interval 8.2–7.5 Ma, the Site U1338 age model was adjusted to that of Site 1146 by tuning the $\delta^{13}C$ records. Lilac shading marks global $\delta^{13}C$ decline coincident with planktic $\delta^{18}O$ increase and high-amplitude obliquity modulation of benthic $\delta^{18}O$. Blue shading marks final stage of global $\delta^{13}C$ decline. Light orange shading marks climate warming after 5.5 Ma. Smooth curve in **b** fitted using locally weighted least squared error (Lowess) method

sediment fraction at Site 1146 [40]. In addition, the predominance of a mollusc group preferring cold-arid conditions in the loess and paleosol layers of central China between 7.1 and 5.5 Ma is indicative of a dominant winter monsoon regime over this period[41].

In contrast to these major hydrological changes in the Northern Hemisphere, mean benthic $\delta^{18}O$ suggests only a

relatively modest, stepwise glacial expansion of the Antarctic ice sheet and/or deep water cooling at ~7 Ma (Figs. 2c and 4f). However, the intensification of the southeast Asian winter monsoon after ~7 Ma was associated with a long-term trend toward heavier benthic $\delta^{18}O$ maxima, which culminated in the most intense maxima (TG22, 20, 14, and 12 between 5.8 and 5.5 Ma) within the entire late Miocene, before reversing in the

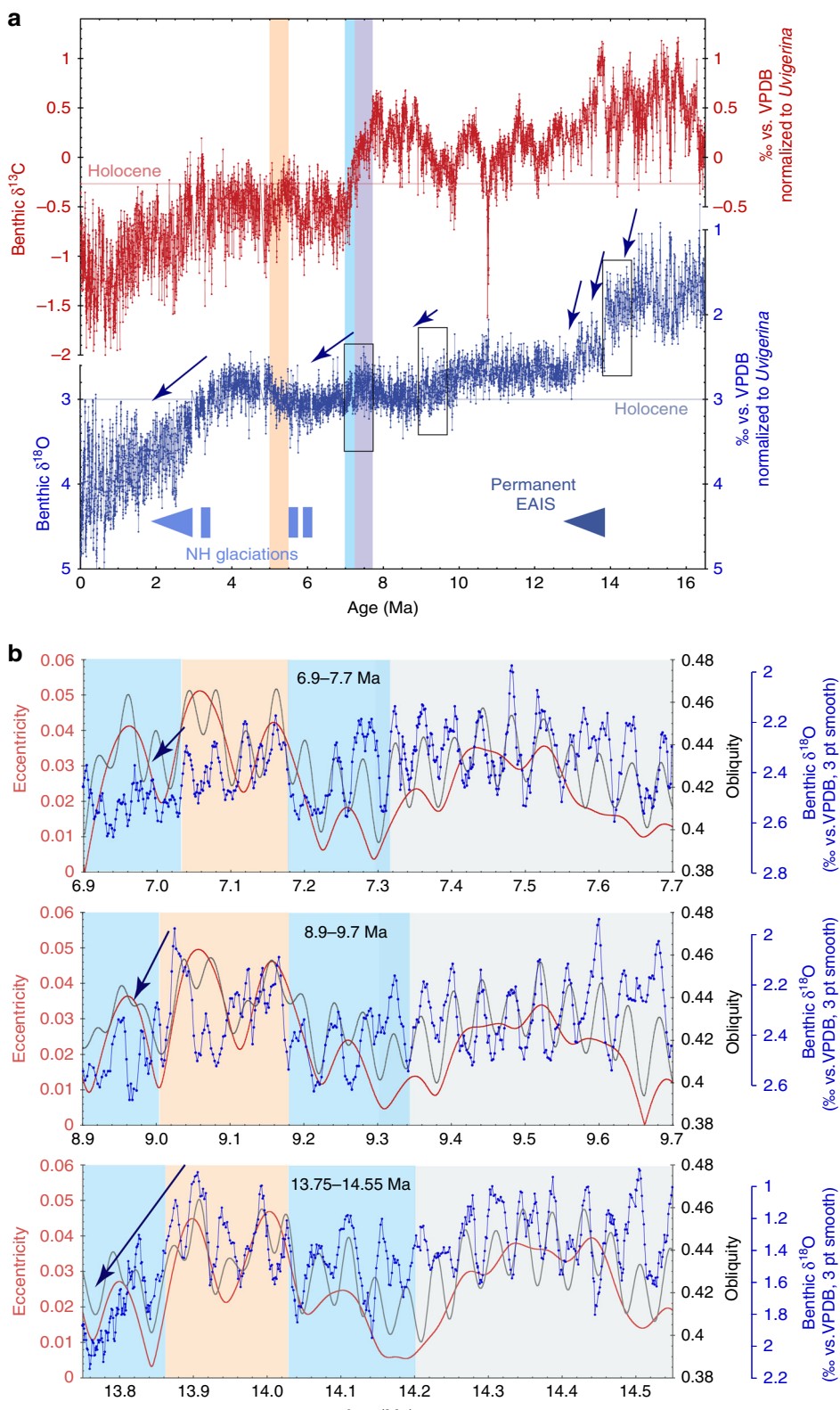

early Pliocene (Figs. 2c and 5f). During these extreme events, benthic $\delta^{18}O$ hovered close to 3‰ (~0.4–0.6‰ increase), which is in the range of late Pliocene values and of intermediate values between peak Holocene and glacial levels at the same location[18]. A previous study[42] related these intense $\delta^{18}O$ maxima to episodes of Antarctic ice ice volume increase. However, the Site 1146 records show that benthic $\delta^{18}O$ maxima (TG22, 20, 14, 12, 4, and T8) coincide with planktic $\delta^{18}O$ maxima between 6.0 and 5.0 Ma, indicating concomitant variations in deep water $\delta^{18}O$ and regional hydrology, which is closely linked to extra-tropical Northern Hemisphere climate variations (Fig. 5e, f). Mixed layer temperatures additionally show concurrent sharp decreases of 2–3 °C during these events (Fig. 5a), implying massive Northern Hemisphere cooling down to subtropical latitudes. The occurrence of ice rafted debris in North Pacific[43] and North Atlantic[44] sediment cores further indicates Northern Hemisphere ice buildups between 6 and 5 Ma. Expansion of Arctic sea ice during these intense cold spells would have increased the positive albedo feedback, amplifying cooling and favoring ice growth. Together these lines of evidence support the development of ephemeral Northern Hemisphere ice sheets (e.g., Greenland, Alaska, Labrador) between 6.0 and 5.5 Ma that were highly susceptible to insolation forcing.

The Site 1146 records additionally reveal that climate cooling and intensification of the winter monsoon at ~7 Ma coincided with the final stage of a long-term, global benthic, and planktic $\delta^{13}C$ decline[27,28] (LMCIS, Figs. 2a and 4c, d). This major shift of ~1‰, which started close to 7.8 Ma, has been interpreted as a global decrease in the $\delta^{13}C$ of the dissolved inorganic carbon pool, although its causes remain debated (e.g., refs. [22,45,46]). A long-held view among contending hypotheses is that this global $\delta^{13}C$ decrease was linked to the late Miocene spreading of C4 grasslands, which are better adapted to low $pCO_2$ and to reduced seasonal precipitation. This large-scale expansion is thought to have resulted in a transfer of $^{13}C$ from the marine to the terrestrial carbon pool[47–49]. A decrease in atmospheric $pCO_2$, linked, for instance, to long-term changes in the oceanic and/or terrestrial carbon inventories, could explain climate cooling after ~7 Ma associated with equatorward migration of the ITCZ and contraction of the WPWP.

The gradient between benthic and planktic $\delta^{13}C$ additionally provides insights into changes in atmospheric $pCO_2$, since it is influenced by two main factors: the sequestration efficiency of the biological pump and equilibration processes between the upper ocean and the atmosphere (Supplementary Note 4; Supplementary Fig. 10). The equilibration time for $\delta^{13}C$ in the mixed surface layer of the ocean exhibits a linear correlation to the ratio of dissolved inorganic carbon to $pCO_2$, which leads to slow equilibration and elevated $\delta^{13}C$ in the mixed layer of the ocean with respect to the atmosphere under low $pCO_2$. Recent model simulations showed that accelerated equilibration under elevated atmospheric $pCO_2$ decreases the isotopic disequilibrium, leads to lower upper ocean $\delta^{13}C$ and, thus, decreases the gradient between the $\delta^{13}C$ of surface and deep water masses[50]. Consequently, the vertical $\delta^{13}C$ gradient in the

ocean exhibits a gentler slope under high atmospheric $pCO_2$ and steepens during intervals of declining $pCO_2$.

Steepening of the gradient between planktic and benthic $\delta^{13}C$ after ~7 Ma at Site 1146, when mixed layer temperatures also declined (Fig. 3a, b), suggests that $pCO_2$ levels decreased, eventually reaching levels that enabled the formation of transient Northern Hemisphere ice sheets between 6.0 and 5.5 Ma. This steeper gradient also denotes a prolonged interval of substantially enhanced marine productivity and accumulation rates of biogenic components ("biogenic bloom" originally described in ref. [51]) at numerous locations in the Pacific, Indian and Atlantic Oceans (e.g., ref. [52] and references therein). In the eastern equatorial Pacific Ocean, opal, and carbonate deposition reached a maximum between 7.0 and 6.4 Ma during the peak of the biogenic bloom in the region[46]. Thus, a plausible scenario is that changes in nutrient supply and/or pathways stimulated marine productivity after ~7 Ma. Steepening of the equator to pole temperature gradient associated with global cooling after ~7 Ma (Fig. 3a, b; ref. [2]) promoted intensification of the Hadley and Walker circulation with repercussions on the wind-driven circulation and precipitation patterns (e.g., ref. [53]). The strengthening of winds may have in turn fostered upwelling and ocean fertilization, helping to drive intense biogenic blooms through the Pacific Ocean, which enhanced carbon storage and decreased $pCO_2$ in the ocean in a positive feedback loop.

Previous work showed that the amplitude of the LMCIS differs in ocean basins (e.g., ref. [54]). In particular, a comparison of benthic $\delta^{13}C$ profiles indicates that the gradient between the Pacific and Atlantic Oceans intensified during the final stage of the LMCIS (Fig. 6b; ref. [54]). The steeper inter-basinal gradient after ~7 Ma cannot be explained by increased production and southward advection of North Atlantic Deep Water, as this relatively warm and/or fresh (lighter $\delta^{18}O$) and $^{13}C$-enriched water mass appears not to have spread into the South Atlantic and Southern Ocean, which remained influenced by colder, denser (heavier $\delta^{18}O$) and $\delta^{13}C$ depleted water masses through the late Miocene (Fig. 6a, b; ref. [54]). Alternatively, the steeper inter-basinal $\delta^{13}C$ gradient after ~7 Ma may be driven by increased export of nutrient enriched waters with a lower preformed $\delta^{13}C$ from the Southern Ocean into the Pacific Ocean (e.g., ref. [54]) and/or to enhanced primary productivity and nutrient regeneration in the low-latitude Pacific Ocean.

Comparison of benthic $\delta^{13}C$ profiles from Site U1338 in the abyssal equatorial Pacific Ocean[55] and the shallower Site 1146 in the northwestern subtropical Pacific Ocean (Fig. 6b) shows that the composition of Pacific water masses changed after 7.2 Ma. The convergence of the $\delta^{13}C$ records after 7.2 Ma indicates expansion of a $\delta^{13}C$ depleted central Pacific deep water mass into shallower depths during the peak of the biogenic bloom. If primarily driven by increased productivity and nutrient regeneration in the Pacific and Indian Oceans, the expansion of a $^{12}C$-enriched deep water mass after 7.2 Ma also implies increased carbon storage in the deep Pacific Ocean. The global efficiency of the biological pump reflects a balance between high- and low-latitude regions with different

**Fig. 7** Middle to late Miocene climate cooling steps coincident with unusual congruence of the Earth's orbit. **a** Miocene to Pleistocene (16–0 Ma) benthic $\delta^{18}O$ and $\delta^{13}C$ records from ODP Site 1146, compiled from refs. [15–18] and this work. Blue arrows mark main phases of glacial expansion/deep water cooling; 3 pt smooth: 3 pt moving mean. Lilac shading marks global $\delta^{13}C$ decline coincident with planktic $\delta^{18}O$ increase and high-amplitude obliquity modulation of benthic $\delta^{18}O$. Blue shading marks final stage of global $\delta^{13}C$ decline. Light orange shading marks climate warming after 5.5 Ma. **b** Comparison of benthic (*C. wuellerstorfi* and/or *C. mundulus*) $\delta^{18}O$ from ODP Site 1146 ([15–17] and this work) with orbital parameters (eccentricity and obliquity from ref. [21]) reveals similar sequence of climatic events during three Miocene cooling episodes with strikingly similar background orbital configuration. Blue shading marks cooling episodes following an extended period of high-amplitude variability in obliquity congruent with low variability in short eccentricity (gray shading). Light orange shading marks transient warming episodes coincident with high-amplitude variability in short eccentricity

sequestration efficiency[56]. Thus, enhanced productivity and organic matter export in the tropical and subtropical ocean may increase global sequestration efficiency and lower atmospheric $pCO_2$, even when deep water formation occurs in high-latitude areas with an inefficient biological pump.

The integrated benthic isotope data in Site 1146 provide the first continuous, highly resolved time series from a single site that span the last 16.4 Myr (Fig. 7a). These extended records track the transition from a warmer middle Miocene climate phase with a reduced and highly dynamic Antarctic ice cover (until ~14 Ma) to an increasingly colder mode with more permanent and stable ice sheets in the late Miocene[17]. These records additionally allow us to evaluate the long-term relationship between radiative forcing and the response of the ocean/climate that is imprinted on the benthic $\delta^{18}O$ signal. For instance, the 41 kyr obliquity cycle is especially prominent in the benthic $\delta^{18}O$ series between 7.7 and 7.2 Ma (Fig. 2c), during a configuration of the Earth's orbit, when high-amplitude variability in obliquity is congruent with extremely low-amplitude variability in short eccentricity (Supplementary Figs. 3, 4A, E). The onset of the ~80 kyr long positive excursion in benthic $\delta^{18}O$ centered at 7.2 Ma notably coincides with minima in obliquity (41 kyr) and in eccentricity (100 kyr, 400 kyr, and 2.4 Myr amplitude modulation) (Fig. 7b; Supplementary Fig. 3). At obliquity and eccentricity minima, lower summer insolation at high latitudes inhibits melting of snow and ice. This conjunction of climatic forcing factors likely fostered a sustained cold phase in the high latitudes that lasted through two consecutive obliquity cycles, resulting in an extended benthic $\delta^{18}O$ positive excursion. Renewed high-amplitude variations in eccentricity and precession together with maximum amplitude variability in obliquity probably drove the successive rebounds between 7.2 and 7.0 Ma.

This sequence of climatic events, as well as their background orbital configuration were strikingly similar during two previous Miocene cooling episodes: the middle Miocene climate transition at ~13.9 Ma, which resulted in major expansion of the East Antarctic ice sheet and the less pronounced late Miocene cooling step at ~9.0 Ma (Fig. 7b). In all three instances, the 41 kyr cycle initially stands out in the benthic $\delta^{18}O$ signal during a protracted period of high-amplitude variability in obliquity, congruent with low variability in short eccentricity. A marked enrichment in benthic $\delta^{18}O$ (0.2–0.3‰), indicative of ice growth and/or deep water cooling toward the end of this interval, coincides with prolonged minima in eccentricity, lasting ~100–200 kyr. Subsequent rebounds at peak insolation, linked to changes in eccentricity cadence (from 400 to 100 kyr variability), indicate episodes of transient ice sheet disintegration and deep water warming. This unusual orbital congruence appears propitious to high-latitude cooling in the Northern and Southern Hemispheres, although boundary conditions differed markedly during these three intervals of climate change. The middle Miocene cooling step occurred in a much warmer climate phase, characterized by substantially lighter mean benthic $\delta^{18}O$ (Fig. 7a, b). At this time, the less extensive ice cover over Antarctica was likely more dynamic and highly responsive to Southern Hemisphere summer insolation[57,58], in contrast to the more expanded Antarctic ice sheet during the late Miocene. This long-term perspective illustrates the nonlinear response of the ocean/climate system to orbital forcing and the role of internal feedback processes including ice sheet hysteresis, latitudinal temperature gradients, ocean circulation and $CO_2$ exchange between terrestrial, atmospheric and oceanic reservoirs.

Arguably, the uncertainty of the $CO_2$ forcing during the Miocene remains a major challenge for defining the characteristics and dynamics of warmer climate states. Although, current $pCO_2$ reconstructions show no significant change through the late Miocene, with levels staying close to or slightly above preindustrial levels, uncertainties in excess of 200 p.p.m. (see compilations in refs. [9,59,60]) preclude assessment of variability and sensitivity to $CO_2$ forcing within the critical preindustrial to modern range. To test the sensitivity of outputs to $pCO_2$ uncertainties, some simulations of late Miocene climate using coupled atmosphere–ocean circulation models have applied atmospheric $pCO_2$ concentrations in the preindustrial range (~280 p.p.m.), as well as more elevated levels of 400–450 p.p.m. (e.g., refs. [60,61]). These studies indicated major changes in vegetation distribution[60] and sea ice cover[61] in the Northern Hemisphere under these different $pCO_2$ states. In particular, forest areas decreased and the albedo of the Eurasian and North American landmasses increased under lower $pCO_2$, due to the markedly lower (by 4–10 °C) mean air temperatures and reduced precipitation during boreal winter[60]. These findings are in agreement with a previous modeling study[62], which found that vegetation changes were more important than paleogeography in determining late Miocene climate. Simulated mean summer SST and sea ice concentrations in the Arctic Ocean[61] also showed that the region is highly sensitive to relatively small changes in $pCO_2$, as a year-round sea ice cover prevails in the central Arctic Ocean at preindustrial levels, whereas summer conditions are ice-free at concentrations of 450 p.p.m. This difference in seasonal ice cover is critical because it implies very different feedbacks in terms of albedo and heat exchange with far-reaching repercussions for global climate[61]. Recent model simulations of Pliocene warmth additionally highlighted the importance of feedbacks associated with cloud albedo and ocean mixing in driving changes in meridional and zonal temperature gradients, despite relatively modest changes in $pCO_2$[63–66].

Data from this study support that subtropical climate cooling and intensification of the southeast Asian winter monsoon after ~7 Ma were synchronous with decreasing $pCO_2$ (Figs. 3a and 4b) within a global context of steepening meridional thermal gradients[2]. We speculate that this late Miocene climate shift was associated with a relatively small decline in $pCO_2$, which was amplified by a conjunction of positive feedbacks. Variations in Northern Hemisphere sea ice cover and vegetation in concert with changes in ocean–atmosphere circulation were likely instrumental in driving late Miocene climate, as illustrated by recent modeling simulations of late Miocene climate[60–62]. The dynamic behavior of the ocean–climate system between 9 and 5 Ma suggests a tight coupling between carbon cycle variations and low-latitude climate evolution. In particular, our results show that changes in Antarctic ice volume were not the primary driver of late Miocene climate development and that low-latitude processes, including monsoonal wind forcing of upper-ocean circulation and productivity had a strong influence on climate-carbon cycle dynamics. Inception of colder climate conditions at ~7 Ma during the final stage of the LMCIS coincided with intensification of the Asian winter monsoon and strengthening of the Pacific Ocean's biological pump, which persisted until ~5.5 Ma. This suggests that changes in the global carbon cycle involved transfer of terrestrial carbon in a cooling, drying climate, as well as fluctuations in the carbon storage capacity of the deep ocean and the sedimentary carbon sink. Ephemeral Northern Hemisphere glaciations between 6.0 and 5.5 Ma additionally indicate that atmospheric $pCO_2$ levels hovered close to and occasionally reached the threshold necessary for Northern Hemisphere ice sheet growth during this period.

## Methods

**Revision of shipboard sediment splice.** Cores were sampled in ~5 cm intervals (~2 kyr time resolution) from a composite sequence (shipboard splice) of Holes 1146A and 1146C (Cores 1146C-30X to 1146C-39X). After comparison of the shipboard natural gamma ray, color reflectance, magnetic susceptibility data, and overlapping benthic isotope records over the splice tie points, we made the following modification to the original shipboard splice: we defined a new tie point between Cores 1146C-38X and 1146A-39X (1146C-38X-4, 122 cm at 359.52 m below sea floor (mbsf) tie to 1146A-39X-2, 107 cm at 359.87 mbsf), based on the match of isotope data from Holes 1146A and C. This adjustment resulted in the addition of an 80 cm splice segment from Hole 1146A to the meter composite depth (mcd) scale.

**Benthic and planktic foraminiferal stable isotopes.** All samples were oven dried at 40 °C and weighed before washing over a 63 μm sieve. Residues were oven dried at 40 °C on a sheet of filter paper, then weighed and sieved into different size fractions. We measured $\delta^{18}O$ and $\delta^{13}C$ in the epifaunal benthic foraminifers *C. wuellerstorfi* and/or *C. mundulus* and on the mixed layer foraminifer *G. sacculifer*. Well-preserved tests were broken into large fragments, cleaned in alcohol in an ultrasonic bath, then dried at 40 °C. Stable isotopes were measured with a Finnigan MAT 253 mass spectrometer at the Leibniz Laboratory, University of Kiel. The instrument is coupled on-line to a Carbo-Kiel Device (Type IV) for automated $CO_2$ preparation from carbonate samples for isotopic analysis. Samples were reacted by individual acid addition (99% $H_3PO_4$ at 75 °C). On the basis of the performance of international and lab-internal standard carbonates, the precision is better than ±0.09‰. Paired measurements in middle Miocene samples from ODP Sites 1146 and 1237 previously indicated no significant offset in $\delta^{18}O$ and $\delta^{13}C$ between *C. wuellerstorfi* and *C. mundulus*[16]. Results were calibrated using the National Institute of Standard and Technology (Gaithersburg, MD) carbonate isotope standard and NBS (National Bureau of Standard) 19 and NBS 20, and are reported on the Vienna PeeDee Belemnite scale.

**Astronomically tuned chronology.** The chronology is based on minimal tuning[67] of the benthic oxygen isotope record to a computed orbital solution[21] (Supplementary Note 1). We used an ET composite with equal weight of eccentricity and obliquity and tuned $\delta^{18}O$ minima to ET maxima (Supplementary Fig. 4). This strategy is based on the notion that relatively warm summers at high obliquity promote ice sheet melting at high latitudes and that a low summer insolation gradient between low and high latitudes at high obliquity leads to a decrease in poleward moisture transport, inhibiting ice sheet buildup[68]. We did not adjust our tuning for possible phase lags between $\delta^{18}O$ and insolation forcing, since the response time of smaller Miocene ice sheets is unknown. We did not include high-latitude summer insolation and/or precession (P) parameters in our tuning target to remain independent of possible changes between dominant northern (ET−P) or southern (ET+P) hemisphere precessional insolation forcing. We note that the obliquity cycle (41 kyr) and the eccentricity cycles (100 and 400 kyr) are also prominently imprinted in the benthic $\delta^{13}C$ record (Supplementary Fig. 6A). The detection of these astronomical frequencies in the $\delta^{13}C$ record supports the age model, based on independent tuning of the $\delta^{18}O$ series. Age correlation points are given in Supplementary Fig. 4D and Supplementary Table 1.

**Mixed layer temperature reconstructions.** Ma/Ca measurements were performed on ~30 well-preserved specimens of *G. sacculifer* from the 250–315 μm fraction (average weights of ~400 μg). Sample spacing is 20 cm from Sample 1146C-30X-1, 33 cm (298.43 revised mcd) to Sample 1146A-39X-4, 88 cm (387.28 revised mcd), corresponding to a mean temporal resolution of ~7 kyr between 8.11 and 5.07 Ma. Sample spacing was decreased to 5 cm, corresponding to a temporal resolution of ~2 kyr, over the prominent benthic $\delta^{18}O$ maxima TG4, TG12, TG14, and TG22 between 6.0 and 5 Ma. Tests of *G. sacculifer* were crushed between glass plates and cleaned with methanol and reductive (hydrazine) and oxidative (hydrogen peroxide) steps, as detailed in refs. [69,70]. Samples were then leached and diluted with nitric acid and analyzed with a SPECTRO Ciros$^{CCD}$ SOP ICP-OES at the ICP-MS Laboratory of the Institute of Geosciences, University of Kiel. We monitored Fe/Ca, Al/Ca, and Mn/Ca ratios to assess the efficacy of the cleaning procedure. These trace-element/Ca ratios do not show significant correlation with Mg/Ca. Mixed layer temperatures were estimated from Mg/Ca ratios using the exponential equation for *G. sacculifer* with an assumed A constant of 0.09[71]. Fifty-four duplicate measurements show a mean Mg/Ca standard deviation of 0.13 mmol mol$^{-1}$. The error associated with mixed layer temperature estimates was defined by propagating the errors introduced by the Mg/Ca measurements and the Mg/Ca temperature calibration assuming no covariance among these errors, following ref. [72].

**Time frequency analysis.** Orbital tuning, bandpass-filtering, calculation of sedimentation rates, and Blackman–Tukey cross-spectral analyses are performed with AnalySeries 2.0.4.2[73]. Cross-wavelet analysis is performed following ref. [74]. We utilize wavelet coherence to quantify coherence and phase relationships between the benthic $\delta^{13}C$ and $\delta^{18}O$ time series in frequency space. We employed the Morlet wavelet and a Monte Carlo count of 300 to establish the statistical significance level (5%). Prior to analysis, both time series were interpolated to 2 kyr time steps and standardized (zero mean and unit standard deviation). Software is available at http://www.pol.ac.uk/home/research/waveletcoherence/.

**Data availability.** Data are archived at the Data Publisher for Earth and Environmental Science (https://doi.pangaea.de/10.1594/PANGAEA.887393).

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

## Acknowledgements

This research used samples provided by the Ocean Drilling Program and was funded by the Deutsche Forschungsgemeinschaft (grants nos. KU649/30-1 and KU649/31-1). K.G. D.D. received a PhD scholarship (CNPq grant no. 244926/2013-1) from Conselho Nacional de Desenvolvimento Científico e Tecnológico, Brazil. We acknowledge financial support by Land Schleswig-Holstein within the funding programme Open Access Publikationsfonds.

## Author contributions

A.E.H. and W.K. conceived the project and wrote the paper. A.E.H., S.C.C., and N.A. generated stable isotope measurements. K.G.D.K., J.J., and J.L. generated Mg/Ca measurements. A.E.H., W.K., and S.C.C. analyzed the data. All authors contributed ideas and discussed the paper.

## Additional information

**Competing interests:** The authors declare no competing interests.

