## [Peer Review File · Nature Communications]

Editorial note: The Figure on page 16 in this Peer Review File is reproduced with permission from Springer Nature: Ma, W., Chai, F., Xiu, P. et al. *Geo-Mar Lett* (2014) 34: 541. <https://doi.org/10.1007/s00367-014-0384-0>

Reviewers' comments:

Reviewer #1 (Remarks to the Author):

Past warm climates enable us to study the functioning of physics, biology, and chemistry of the Earth System under conditions that might be similar to the near future, so if a study is sufficiently well-done it can be of broad and immediate importance. The late Miocene is a period of substantial interest in this regard, with atmospheric CO₂ conditions, which have large uncertainties, but which are generally in the modern range and yet with temperature conditions much warmer and with reduced gradients compared to modern. The current effort by Holbourn et al has several strengths and important weaknesses.

The multiple time series that the authors have put together are impressive in terms of their very fine temporal resolution and the fact that multiple proxies are brought to bear. This required a huge effort clearly. The quality of the interpretation of the records leaves much to be desired however. The presentation of the data itself is excellent, but most of the paper is given over to many pages of rampant speculation on causes, discussion, and implications. This is mostly hand-waving, with little that is actually quantitatively testable and no quantitative comparisons are given with prior data or modeling results which might provide some context.

Many of the largest claims in the paper are dramatic overstatements given the limitations of the records.

For example "Our high-resolution climate proxy data reveal that high-latitude cooling at 7.2-7.0 Ma was synchronous with a re-organization of subtropical climate." Very little evidence is provided for a "reorganization of subtropical climate". How could one show that with only one site? It would take dozens of sites to make such claims.

Without a quantitative compilation of all existing paleotemperature/paleo-salinity records mapped spatially and interpreted through the lens of atmosphere-dynamics provided by climate modeling, this description is unsupported (other papers, for example by Fedorov and Ravelo, have attempted this).

The contention that the results show an "intensification of the Asian winter monsoon" seems weak and again unsupported in a quantitative fashion. The statement that there is a "strengthening of the biological pump until ~5.5 Ma" seems better supported by the $\delta^{13}\text{C}$ records and is more directly attributable without modeling. The direct but hand-waving relationship between "global $\delta^{13}\text{C}$ records" and "atmospheric pCO₂" which "drove the ocean-climate system over a critical threshold into a new mode of variability" are just conjectures. A better approach would actually reconstruct pCO₂ either in a proxy sense or in an inverse model sense (using GEOCARB etc), the current approach is neither accurate or useful.

In essence, this paper by itself would make a decent Paleoclimatology paper, since clearly the data are of high quality and should be published. But, there is not enough strong linkage between the data and the huge conjectures made about the climate system to make this convincing or interesting enough for a Nature Communications paper.

Reviewer #2 (Remarks to the Author):

I was very excited to read this paper, as the authors have produced some of the most compelling data sets and analysis of Miocene paleoclimatology in previous, highly cited work. The present data set builds on long existing Miocene time series from Site 1146- adding to what may be the most highly resolved set of paleoclimate records from one location for this time period.

What's new? A strong high-low latitude coupling (cooling trend) in late Miocene time was already proposed by Herbert et al. 2016 based on surface ocean temperature estimates, but here the temporal resolution is at least an order of magnitude better. The claim in the first paragraph as to the paucity of well resolved late Miocene climate time series is accurate. I was particularly interested to see stable isotopic time series from planktonic foraminifera, as nearly all the isotopic data published deals with benthic foraminifera. The planktonic data do in fact provide a fresh perspective to late Miocene climate evolution in my opinion.

However, there is one recent paper that is conspicuously absent- Drury et al. (2016) which also presents a detailed analysis of highly resolved isotope records from another deep ocean site. That paper makes claims about carbon cycle-ice sheet coupling over this time that would be very important for the authors to evaluate- more about this later.

As an overview, I applaud the authors for what will surely be an influential work, but I do not think the paper is strong enough in present form. In essence, the paper strains too hard to come up with simple interpretations- a fundamental step at a particular time period (7.2 Ma) and a reorganization of climate- that are not supported by rigorous data analysis, and, that in my opinion, try to simplify a much more evolutive and mysterious transit in climate history prematurely. There are times where we may be more effective in highlighting ambiguities or apparent mismatches in data sets than in attempting clear answers. In our 2016 paper, for example, we pointed out the lack of a strong oxygen isotopic trend at the same time that surface temperatures over most of the earth cooled strongly. To me, this suggests that while continental ice volume dominates the benthic $\delta^{18}\text{O}$ record, ice volume does not necessarily dominate global climate. Our paleoclimatic world view has relied so strongly on the $\delta^{18}\text{O}$ proxy that we need to open ourselves to the story that other proxies have to tell about the tropics, about heat transport on the poleward side of the gyre, the global carbon cycle, etc.

I would like to see interpretations based less on visual (and potentially fairly subjective) analysis and more on rigorous statistics. As I read through the text, I just didn't find a compelling analysis to support the proposed increased coupling of high and low latitude climate after 7.2 Ma. As an example of what seems like loose analysis, the interval of time interpreted as "dynamic EAIS" from 9-7.7 Ma in Figure 2 has $\sim\sim$ same value of benthic $\delta^{18}\text{O}$ as "expanded, more stable EAIS" from 6.3-7 Ma. Where is the evidence for any fundamental shift in behavior of the EAIS? The benthic $\delta^{18}\text{O}$ trend prior to 7.2 Ma is quite different from the planktonic trend. Surely, these ambiguities are intriguing and worth thinking about.

It seemed to me that some of the most compelling similarities in the data set come from looking at carbon isotope information. First, the coherency of the benthic and planktonic $\delta^{13}\text{C}$ time series seems far higher than for $\delta^{18}\text{O}$, and this is worth thinking about. Second, the $\delta^{13}\text{C}$ series is much more similar to the planktonic $\delta^{18}\text{O}$ data set, in that it shares the long wavelength trends, particularly before 7.2 Ma. This would couple $\delta^{13}\text{C}$, planktonic $\delta^{18}\text{O}$, and our alkenone SST estimates into a coherent picture of surface ocean cooling coupled to long-term carbon cycle change. The long term carbon cycle is mentioned in the text, but I think it could be emphasized differently to highlight similarities to the planktonic $\delta^{18}\text{O}$ record. In addition, it looks to me as if the $\delta^{13}\text{C}$ record becomes much more coherent with the benthic $\delta^{18}\text{O}$ record after ~ 7.2 Ma, which might support the contention that the system evolved significantly after that time (I attach 2 figures where I superimposed the benthic $\delta^{13}\text{C}$ onto first the planktic, and then the benthic $\delta^{18}\text{O}$ curves). A unified theory might be that the long term trend represents transfer of terrestrial carbon in a cooling, drying climate on land, which is mentioned in the text but could be highlighted. Moving correlations or moving coherency spectral estimates based comparing the $\delta^{13}\text{C}$ records with planktonic and benthic $\delta^{18}\text{O}$ might be very revealing.

I have attached a figure modified from the authors that projects the $\delta^{13}\text{C}$ data onto planktic and benthic $\delta^{18}\text{O}$ to make my point.

The big problem to me with the present emphasis on identifying a discrete step and change to a "new equilibrium" post-7.2 Ma is that climate change on the Myr time scale plays out through the medium of modulations and amplifications of the orbital-scale cyclicity in all proxy records. This means that in most cases there is a wide range of frequencies expressed in climatic time series, and drawing lines to indicate shifts in system behavior is often artificial. Paleoclimatic data do show occasionally show regime shifts, but in my opinion such instances are in fact rare when analyzed rigorously rather than qualitatively. I also do not see evidence for climate after 7.2 Ma following "equilibrium" behavior- how would such a state make itself clear in the data sets presented? I would avoid the certainty implied by a vocabulary of system behavior until a point where that behavior can be demonstrated numerically.

In a similar vein, although CO₂ control is invoked many times, the case is still circumstantial, as proxy reconstructions are both of low temporal resolution and questionable accuracy. So the present data cannot be used to claim cause and effect, although they are suggestive of an overarching CO₂ guidance of paleoclimate transitions in the late Miocene.

It seems to me that the excellent figures included in this paper argue for a different tack- a more evolutive understanding of late Miocene climate, and perhaps a focus on the transit to the latest Miocene glacial events?

Areas of proxy interpretation that can be improved include the controls on planktonic $\delta^{18}\text{O}$ and Mg/Ca. Can the authors discuss more the possibility of regional hydrological signals in the planktonic record? How do we know how important that overprint could be? Also, I find it hard to take the "absolute" temperature estimates of the Mg/Ca analyses at face value. They would make most of the record as cold or colder than winter temperatures today. Everything we know about Miocene climate (to ~6 Ma) says that it was warmer than today. Alkenone-derived temperatures at Site 1146 become saturated ($> 28.5^\circ\text{C}$) by the middle Pliocene (Li et al., 2011). Furthermore, although it is controversial, the Zhang et al TEX86 data from the Western Pacific warm pool support a regional warming of several degrees relative to today in the late Miocene. If the Mg/Ca temperature estimates were revised upward, there would be significant consequences for the inferred $\delta^{18}\text{O}_{\text{sw}}$ presented in Supplementary Figure 8. The authors do not have to accept my opinion of SST reconstructions in order to at least acknowledge the potential for Mg/Ca estimates to suffer a bias, based on other proxies.

A few minor points. In the Supplement- the wavelet spectral analyses appear look to me like the benthic $\delta^{18}\text{O}$ record has more consistent 100 kyr power than the planktonic, which doesn't make particular sense if 41 kyr cycles dominate i.v. as the authors claim. More generally, it's not evident that the wavelet approach backs up the label on Fig 2 of main text for 7.2-7.7 Ma as "obliquity dominated".

It's interesting that the standard deviation of Planktic $\delta^{18}\text{O}$ shows much more evolution over time than benthic.

Please see specific comments annotated on the .pdf of the main text

Reviewer #3 (Remarks to the Author):

Holbourn et al. have constructed orbital-scale stable isotope and Mg/Ca records from ODP Site 1146, South China Sea, spanning the late Miocene (9-5 Ma). The focus of the study is a warming event between 7.7-7.2 Ma followed by a cooling event 7.2-7.0 Ma, which is to mark a major climate re-organization (e.g., line 295). The data are of high quality, and generally the manuscript is well written.

What the authors refer to as a 'major' cooling event is exhibited as a 0.3 per mil increase in benthic foram d18O values (~1 degree C) and a 2 degree C cooling of mixed layer temperatures. Thus in comparison to major events such as at the Eocene/Oligocene boundary, the mid Miocene, or the onset of significant Northern Hemisphere Glaciation, this one seems relatively minor. But this does not mean that it is not worth looking at in greater detail.

The authors argue for 'substantially higher amplitude variability' (line 106) of the climate signal after the cooling event at 7.2-7.0 Ma. I suggest to show this more quantitatively using time series analysis. At the very least, they should mention whether or not the difference in the variability is statistically significant (lines 106/107).

I don't think that planktonic-benthic foraminiferal d13C gradients are a commonly accepted proxy for atmospheric CO2 levels (Figure 3A) (discussion beginning on line 200). Although the authors make an argument about how it works conceptually, I think in reality the interpretation of the gradient is more complicated than stated. For example, theoretically, the gradient is also reduced when benthic foraminiferal d13C values increase, which can be due to processes in the source region of deep waters and variations in the relative proportions of deep water masses. How does a planktonic-benthic d13C record for the past 800,000 years compare with the ice core pCO2 record? I think the text needs a more thorough discussion of all the factors that can affect the vertical carbon isotope gradient as well as proof of concept (supplemental maybe?).

There are several sentences throughout the text that seem to make strong statements (e.g., line 222 and line 262, see below), but the underlying evidence, or at the very least citations, are not provided.

Figure 4 is only mentioned in passing on lines 137 and 149. I think it shows a lot of information and needs a more thorough description with reference to the specific panels.

In order:

Line 56-59: The sentence is awkward. In essence it means that cooling can be a response to temperature...

Line 222-225: What is the evidence for this statement? I think this needs citations to specific articles.

Line 236 and again line 246: The author use terms that imply statistical analyses (covariance, in phase..), but none are provided, at least not in this context?

Line 260: 'minima in obliquity': I think they mean minima in the amplitude of obliquity variations (a node)?

Line 261-263: How exactly does this work? I think the idea that glaciations occur during times of obliquity nodes has been discussed in the literature (Hilgren's work?).

Line 279: Given Cenozoic ice-volume history, I am not sure that a 0.2- to 0.3 per mil increase in d18O indicates 'substantial' growth of ice and or deep water cooling. I would suggest eliminating qualifying words like 'major', 'substantial' or 'striking'.

We thank the reviewers for detailed, constructive comments that helped us to improve the ms.

Reviewer #1 (Remarks to the Author):

Past warm climates enable us to study the functioning of physics, biology, and chemistry of the Earth System under conditions that might be similar to the near future, so if a study is sufficiently well-done it can be of broad and immediate importance. The late Miocene is a period of substantial interest in this regard, with atmospheric CO₂ conditions, which have large uncertainties, but which are generally in the modern range and yet with temperature conditions much warmer and with reduced gradients compared to modern. The current effort by Holbourn et al has several strengths and important weaknesses.

The multiple time series that the authors have put together are impressive in terms of their very fine temporal resolution and the fact that multiple proxies are brought to bear. This required a huge effort clearly. The quality of the interpretation of the records leaves much to be desired however. The presentation of the data itself is excellent, but most of the paper is given over to many pages of rampant speculation on causes, discussion, and implications. This is mostly hand-waving, with little that is actually quantitatively testable and no quantitative comparisons are given with prior data or modeling results which might provide some context. Many of the largest claims in the paper are dramatic overstatements given the limitations of the records.

For example "Our high-resolution climate proxy data reveal that high-latitude cooling at 7.2-7.0 Ma was synchronous with a re-organization of subtropical climate." Very little evidence is provided for a "reorganization of subtropical climate". How could one show that with only one site? It would take dozens of sites to make such claims.

Without a quantitative compilation of all existing paleotemperature/paleo-salinity records mapped spatially and interpreted through the lens of atmosphere-dynamics provided by climate modeling, this description is unsupported (other papers, for example by Fedorov and Ravelo, have attempted this).

To address the reviewer's criticisms:

1) We provided an additional supplementary figure (Suppl. Fig. 9), which shows a comparison of the 1146 high resolution Mg/Ca based temperature record to the most recent available low to intermediate resolution (mainly based on Uk37) temperature records from ODP and IODP sites in the Arabian Sea, North, East and South Pacific Ocean, North Atlantic Ocean, Norwegian Sea and from Mediterranean land sections (Herbert et al., 2016; Tzanova et al., 2015). Except for the Arabian Sea, all these records (despite the low resolution and poorly constrained stratigraphy at some locations) exhibit a distinct temperature decline starting at ~7.2 Ma, reaching a SST minimum at ~5.5 Ma, which is in agreement with the 1146 subtropical West Pacific record. This comparison demonstrates that the climate cooling registered at Site 1146 is not just a regional feature, but represents a global event associated with a large-scale re-organization of ocean-atmosphere circulation patterns.

2) We thoroughly revised our original interpretations and overhauled the main text, toning down speculative interpretations (see highlighted text). In particular, we revised our initial interpretation of a simple transitional step into a new climate mode at 7 Ma and we now argue for a more evolutive climate transit through the late Miocene, associated with changes in the terrestrial and marine carbon cycle (see revised title, Lines 20-23 in abstract, Discussion, Lines 329-342 in final paragraph).

3) We provided additional time series and statistical analyses (Blackman-Tukey Power Spectra, coherence and phase analyses, locally weighted least squared error, moving window correlation; see Suppl. Figs. 3-4, 14-15), which allow a more rigorous quantitative evaluation of data and interpretations. We refer to these additional results in relevant parts of the main text (see in particular Lines 97-172 in Results, 270-276 in Discussion) and in the Supplement text.

Herbert, T. D., Lawrence, K. T., Tzanova, A., Peterson, L. C., Caballero-Gill, R., Kelly, C. S. Late Miocene global cooling and the rise of modern ecosystems. *Nat. Geosci.* **9**, 843-847 (2016).

Tzanova, A., Herbert, T. D., Peterson, L. Cooling Mediterranean Sea surface temperatures during the Late Miocene provide a climate context for evolutionary transitions in Africa and Eurasia. *Earth Planet. Sci. Lett.* **419**, 71-80 (2015).

The contention that the results show an "intensification of the Asian winter monsoon" seems weak and again unsupported in a quantitative fashion.

We now provide additional information, and references to back up our interpretation of hydrological change associated with intensification of the winter monsoon after 7 Ma. In particular, we show that mixed layer cooling between 7.1 and 5.7 Ma concurs with a long-term increase in the mean and amplitude variability of seawater $\delta^{18}\text{O}$ (Lines 163-166 in Results, Lines 182-198 in Discussion; Lines 156-174 in Supplementary Note 1; Suppl. Fig. 10). These trends signal a change in the $\delta^{18}\text{O}$ composition of precipitation and/or runoff, likely associated with changes in the provenance and/or seasonality of precipitation towards a more pronounced monsoonal seasonality of precipitation and a more temperature controlled seasonality of rainwater $\delta^{18}\text{O}$ (i.e. $\delta^{18}\text{O}$ depleted winter precipitation, Liu et al., 2014).

Intensification of the Asian winter monsoon after 7 Ma is supported by several studies, which we now refer to on Lines 196-201 (Guo et al., 2002, Wan et al., 2007; Li et al., 2008). In particular, Li et al. (2008) analysed molluscs in the loess and paleosol layers in central China, discriminating cold-arid and warm-humid ecological groups, related to dominance of winter (cold-arid) or summer (warm-humid) monsoon conditions, as previously defined for Quaternary glacial–interglacial cycles (e.g., Rousseau and Wu, 1997; Wu et al., 2001). According to Li et al. (2008), variations in these two groups of molluscs indicate dominance of the East Asian winter monsoon between 7.1 and 5.5 Ma. Climate modeling additionally suggests strengthening of the East Asian winter monsoon due to intensified westerlies during the late Miocene (Tang et al., 2011).

Guo et al. Onset of Asian desertification by 22 Myr ago inferred from loess deposits in China.

Nature **416**, 159-163 (2002).

Li, F. J., Rousseau, D. D., Wu, N. Q., Hao, Q. Z., Pei, Y. P. Late Neogene evolution of the East Asian monsoon revealed by terrestrial mollusk record in Western Chinese Loess Plateau: from winter to summer dominated sub-regime, *Earth Planet. Sc. Lett.*, **274**, 439-447 (2008).

Liu, J., Song, X., Yuan, G., Sun, X., Yang, L. Stable isotopic compositions of precipitation in China, *Tellus B: Chemical and Physical Meteorology* **66**:1, 22567, doi:10.3402/tellusb.v66.22567 (2014).

Rousseau, D.D., Wu, N.Q. A new molluscan record of the monsoon variability over the past 130 000 yr in the Luochuan loess sequence, China. *Geology* **25**, 275–278 (1997).

Tang H., Micheels A., Eronen J., Fortelius M. Regional climate model experiments to investigate the Asian monsoon in the Late Miocene. *Clim Past* **7**(3), 847–868 (2011).

Wan S., Li, A., Clift, P. D., Stuu, J.-B. W. Development of the East Asian monsoon: Mineralogical and sedimentologic records in the northern South China Sea since 20 Ma. *Palaeogeogr. Palaeoclim. Palaeoecol.* **254**, 561-582 (2007).

Wu, N.Q., Rousseau, D.D., Liu, T.S., Lu, H.Y., Gu, Z.Y., Guo, Z.T., Jiang, W.Y. Orbital forcing of terrestrial mollusks and climatic changes from the Loess Plateau of China during the past 350 ka. *J. Geophys. Res.* **106**, 20045–20054 (2001).

The statement that there is a "strengthening of the biological pump until ~5.5 Ma" seems better supported by the $\delta^{13}\text{C}$ records and is more directly attributable without modeling. The direct but hand-waving relationship between "global $\delta^{13}\text{C}$ records" and "atmospheric pCO_2 " which "drove the ocean-climate system over a critical threshold into a new mode of variability" are just conjectures. A better approach would actually reconstruct pCO_2 either in a proxy sense or in an inverse model sense (using GEOCARB etc), the current approach is neither accurate or useful.

We revised our original interpretation and removed the criticized statement. We no longer argue that these changes are linked to a discrete climatic step at 7 Ma; instead, we emphasize a more complex, evolutive climate transit through the late Miocene associated with changes in the global carbon cycle (see revised title, Lines 20-23 in abstract, Lines 328-338 in final paragraph). To support our interpretation of the $\Delta\text{d}^{13}\text{C}$ proxy record over the late Miocene at Site 1146, we additionally provided a new figure (Suppl. Fig. 12), which shows that $\Delta\text{d}^{13}\text{C}$ at Site 1146 does reflect pCO_2 variability on a glacial-interglacial scale between 0 and 600 ka. We also added a more detailed discussion of factors influencing the $\Delta\text{d}^{13}\text{C}$ record at Site 1146 in Supplementary Note 2.

We agree that reconstructing late Miocene pCO_2 would be extremely useful. However, we feel that this is beyond the scope of the present study. Developing reliable, high-resolution isotope records and temperature reconstructions that extend over several million years requires several years of dedicated work. We feel that these unique, high-quality data can contribute to a better understanding of Miocene climate evolution, independently from modeling and pCO_2 reconstruction, which are still highly uncertain for the late Miocene.

Reviewer #2 (Remarks to the Author):

I was very excited to read this paper, as the authors have produced some of the most compelling data sets and analysis of Miocene paleoclimatology in previous, highly cited work. The present data set builds on long existing Miocene time series from Site 1146-adding to what may be the most highly resolved set of paleoclimate records from one location for this time period.

What's new? A strong high-low latitude coupling (cooling trend) in late Miocene time was already proposed by Herbert et al. 2016 based on surface ocean temperature estimates, but here the temporal resolution is at least an order of magnitude better. The claim in the first paragraph as to the paucity of well resolved late Miocene climate time series is accurate. I was particularly interested to see stable isotopic time series from planktonic foraminifera, as nearly all the isotopic data published deals with benthic foraminifera. The planktonic data do in fact provide a fresh perspective to late Miocene climate evolution in my opinion.

However, there is one recent paper that is conspicuously absent- Drury et al. (2016) which also presents a detailed analysis of highly resolved isotope records from another deep ocean site. That paper makes claims about carbon cycle-ice sheet coupling over this time that would be very important for the authors to evaluate- more about this later.

In answer to the reviewer's concern, we added a new figure (Suppl. Fig. 11), which shows a comparison of the 1146 benthic record with the benthic data from Drury et al. (2016). This comparison supports that the 1146 benthic isotope signals ($\delta^{18}\text{O}$ and $\delta^{13}\text{C}$) are overall representative of deep Pacific water masses. However, divergence prior to 7.2 Ma (lower benthic $\delta^{13}\text{C}$ in the deep equatorial Pacific Ocean) points to different water mass characteristics prior to the Biogenic Bloom. Convergence of the $\delta^{13}\text{C}$ records after 7.2 Ma suggests expansion of a $\delta^{13}\text{C}$ depleted central Pacific deep water mass into shallower depths, indicating that the composition of deep water masses changed after 7.2 Ma. The expansion of a ^{12}C -enriched deep water mass would also imply increased carbon storage in the deep Pacific Ocean after 7.2 Ma, indicating that changes in deep water circulation influenced ocean carbon storage. This point is now referred on Lines 232-238.

As an overview, I applaud the authors for what will surely be an influential work, but I do not think the paper is strong enough in present form. In essence, the paper strains too hard to come up with simple interpretations- a fundamental step at a particular time period (7.2 Ma) and a reorganization of climate- that are not supported by rigorous data analysis, and, that in my opinion, try to simplify a much more evolutive and mysterious transit in climate history prematurely. There are times where we may be more effective in highlighting ambiguities or apparent mismatches in data sets than in attempting clear answers. In our 2016 paper, for example, we pointed out the lack of a strong oxygen isotopic trend at the same time that surface temperatures over most of the earth cooled

strongly. To me, this suggests that while continental ice volume dominates the benthic $\delta^{18}\text{O}$ record, ice volume does not necessarily dominate global climate. Our paleoclimatic world view has relied so strongly on the $\delta^{18}\text{O}$ proxy that we need to open ourselves to the story that other proxies have to tell about the tropics, about heat transport on the poleward side of the gyre, the global carbon cycle, etc.

To address this criticism, we thoroughly overhauled the main text, including new time series analyses and statistical results (see response to next comment) to better support the findings discussed (Lines 97-172 in Results, 270-276 in Discussion and Supplement text). We also revised our original interpretations of a simple transitional step into a new climate mode at 7 Ma. We agree with the reviewer that statistical analysis supports a more evolutive climate transit through the late Miocene associated with changes in the global carbon cycle.

I would like to see interpretations based less on visual (and potentially fairly subjective) analysis and more on rigorous statistics. As I read through the text, I just didn't find a compelling analysis to support the proposed increased coupling of high and low latitude climate after 7.2 Ma. As an example of what seems like loose analysis, the interval of time interpreted as "dynamic EAIS" from 9-7.7 Ma in Figure 2 has $\sim\sim$ same value of benthic $\delta^{18}\text{O}$ as "expanded, more stable EAIS" from 6.3-7 Ma. Where is the evidence for any fundamental shift in behavior of the EAIS?

We now provide additional statistical analyses of the 1146 data sets (Blackman-Tukey Power Spectra, coherence and phase analyses, locally weighted least squared error, moving window correlation; see Suppl. Figs. 3-4, 14-15; see Suppl. Figs. 3-4, 14-15), which allow a more rigorous quantitative evaluation of data and interpretations. The means of benthic $\delta^{18}\text{O}$ are indeed quite close (2.47 between 6.5 and 7.0 Ma and 2.42 between 7.5 and 9 Ma). However, the comparison of means is strongly dependent on the interval chosen. Within the relatively short interval 6.5 to 7.0 Ma (271 data points) there are two important peaks of light $\delta^{18}\text{O}$, skewing the data towards low values (skewness of -0.75), whereas in the larger dataset 7.5-9.0 Ma (835 data points) the distribution of light and heavy values is more even (skewness of -0.20). Nevertheless, we agree that we previously over-interpreted the record and have now revised this.

The benthic $\delta^{18}\text{O}$ trend prior to 7.2 Ma is quite different from the planktonic trend. Surely, these ambiguities are intriguing and worth thinking about.

It seemed to me that some of the most compelling similarities in the data set come from looking at carbon isotope information. First, the coherency of the benthic and planktonic $\delta^{13}\text{C}$ time series seems far higher than for $\delta^{18}\text{O}$, and this is worth thinking about.

Second, the $\delta^{13}\text{C}$ series is much more similar to the planktonic $\delta^{18}\text{O}$ data set, in that it shares the long wavelength trends, particularly before 7.2 Ma. This would couple $\delta^{13}\text{C}$, planktonic $\delta^{18}\text{O}$, and our alkenone SST estimates into a coherent picture of surface ocean cooling coupled to long-term carbon cycle change. The long term carbon cycle is

mentioned in the text, but I think it could be emphasized differently to highlight similarities to the planktonic $\delta^{18}\text{O}$ record.

In addition, it looks to me as if the $\delta^{13}\text{C}$ record becomes much more coherent with the benthic $\delta^{18}\text{O}$ record after ~ 7.2 Ma, which might support the contention that the system evolved significantly after that time (I attach 2 figures where I superimposed the benthic $\delta^{13}\text{C}$ onto first the planktonic, and then the benthic $\delta^{18}\text{O}$ curves).

We now include a new supplementary figure, which provides a quantitative evaluation of long-term trends (Suppl. Fig. 4A) and a comparison of Site 1146 planktonic and benthic $\delta^{18}\text{O}$ and $\delta^{13}\text{C}$ (Suppl. Fig. 4B) and refer to this new figure on Lines 97-106 in Results. We revised our original interpretations that ice volume was the driver of global climate during the late Miocene and now emphasize that low latitude processes (such as monsoonal winds, large-scale ocean circulation and productivity) also play a fundamental role in regulating $p\text{CO}_2$ and defining climate trends (see Discussion). These findings are summarized in the abstract and final paragraph.

A unified theory might be that the long term trend represents transfer of terrestrial carbon in a cooling, drying climate on land, which is mentioned in the text but could be highlighted. Moving correlations or moving coherency spectral estimates based comparing the $\delta^{13}\text{C}$ records with planktonic and benthic $\delta^{18}\text{O}$ might be very revealing. I have attached a figure modified from the authors that projects the $\delta^{13}\text{C}$ data onto planktonic and benthic $\delta^{18}\text{O}$ to make my point.

We added 2 new figures (Suppl. Figs. 14, 15A and 15B), which show 10 pt window correlation and spectral analysis of coherence and phase in 1 Myr steps. We also included a Supplementary Note (Note 3) that discusses changes in phase relationship between planktonic $\delta^{13}\text{C}$ and $\delta^{18}\text{O}$. We additionally modified the abstract and final paragraph to emphasize the importance of long-term changes in the carbon cycle in driving late Miocene climate evolution (Lines 20-23 and 328-342).

The big problem to me with the present emphasis on identifying a discrete step and change to a “new equilibrium” post-7.2 Ma is that climate change on the Myr time scale plays out through the medium of modulations and amplifications of the orbital-scale cyclicity in all proxy records. This means that in most cases there is a wide range of frequencies expressed in climatic time series, and drawing lines to indicate shifts in system behavior is often artificial. Paleoclimatic data do occasionally show regime shifts, but in my opinion such instances are in fact rare when analyzed rigorously rather than qualitatively. I also do not see evidence for climate after 7.2 Ma following “equilibrium” behavior- how would such a state make itself clear in the data sets presented? I would avoid the certainty implied by a vocabulary of system behavior until a point where that behavior can be demonstrated numerically.

We thoroughly revised our original interpretation of a discrete step at 7-7.2 Ma into a new climate mode and modified the text accordingly. In the revised text, we removed

terms implying shifts in system behavior, in particular we no longer use the term “equilibrium”. Instead of the strong emphasis on, admittedly speculative, identification of regime shifts in system behavior, we now focus the discussion on characteristic short-term climate events and on the long-term evolutive climate transit, revealed by our high-resolution proxy time series (as documented in Figs. 2-5).

In a similar vein, although CO₂ control is invoked many times, the case is still circumstantial, as proxy reconstructions are both of low temporal resolution and questionable accuracy. So the present data cannot be used to claim cause and effect, although they are suggestive of an overarching CO₂ guidance of paleoclimate transitions in the late Miocene. It seems to me that the excellent figures included in this paper argue for a different tack- a more evolutive understanding of late Miocene climate, and perhaps a focus on the transit to the latest Miocene glacial events?

We agree with the reviewer’s comment and toned down our original interpretation of pCO₂ decrease as a trigger of a discrete climate step. Instead, we emphasize the long-term climatic evolution through the late Miocene, culminating with transient NH glaciations (TG events). The discussion is now focused on the relationship between low latitude climate processes (atmospheric and ocean circulation) and changes in the global carbon cycle rather than a distinct climate step (see revised title, Lines 20-23 in abstract, Lines 328-342 in final paragraph).

To support our interpretation of the $\Delta d^{13}C$ proxy record over the late Miocene at Site 1146, we additionally provided a new figure (Suppl. Fig. 12), which shows that $\Delta d^{13}C$ at Site 1146 does reflect pCO₂ variability on a glacial-interglacial scale between 0 and 600 ka. We also added a more detailed discussion of factors influencing the $\Delta d^{13}C$ record at Site 1146 in Supplementary Note 2.

Areas of proxy interpretation that can be improved include the controls on planktonic $\delta^{18}O$ and Mg/Ca. Can the authors discuss more the possibility of regional hydrological signals in the planktonic record? How do we know how important that overprint could be? Also, I find it hard to take the “absolute” temperature estimates of the Mg/Ca analyses at face value. They would make most of the record as cold or colder than winter temperatures today. Everything we know about Miocene climate (to ~6 Ma) says that it was warmer than today. Alkenone-derived temperatures at Site 1146 become saturated (> 28.5°C) by the middle Pliocene (Li et al., 2011). Furthermore, although it is controversial, the Zhang et al TEX86 data from the Western Pacific warm pool support a regional warming of several degrees relative to today in the late Miocene. If the Mg/Ca temperature estimates were revised upward, there would be significant consequences for the inferred $\delta^{18}O_{sw}$ presented in Supplementary Figure 8. The authors do not have to accept my opinion of SST reconstructions in order to at least acknowledge the potential for Mg/Ca estimates to suffer a bias, based on other proxies.

We now provide additional information, and references to back up our interpretation of

hydrological change associated with intensification of the winter monsoon after 7 Ma. In particular, we show that mixed layer cooling between 7.1 and 5.7 Ma concurs with a long-term increase in the mean and amplitude variability of seawater $\delta^{18}\text{O}$ (Lines 163-166 in Results, Lines 182-198 in Discussion; Lines 156-174 in Supplementary Note 1; Suppl. Fig. 10). These trends signal a change in the $\delta^{18}\text{O}$ composition of precipitation and/or runoff, likely associated with changes in the provenance and/or seasonality of precipitation towards a more pronounced monsoonal seasonality of precipitation and a more temperature controlled seasonality of rainwater $\delta^{18}\text{O}$ (i.e. $\delta^{18}\text{O}$ depleted winter precipitation, Liu et al., 2014).

Intensification of the Asian winter monsoon after 7 Ma is supported by several studies, which we now refer to on Lines 196-201 (Guo et al., 2002, Wan et al., 2007; Li et al., 2008). In particular, Li et al. (2008) analysed molluscs in the loess and paleosol layers in central China, discriminating cold-arid and warm-humid ecological groups, which were related to dominance of winter (cold-arid) or summer (warm-humid) monsoon conditions, as previously defined for Quaternary glacial–interglacial cycles (e.g., Rousseau and Wu, 1997; Wu et al., 2001). According to Li et al. (2008), variations in these two groups of molluscs indicate dominance of the East Asian winter monsoon between 7.1 and 5.5 Ma. Climate modeling additionally suggests strengthening of the East Asian winter monsoon due to intensified westerlies during the late Miocene (Tang et al., 2011).

Guo et al. Onset of Asian desertification by 22 Myr ago inferred from loess deposits in China. *Nature* **416**, 159-163 (2002).

Li, F. J., Rousseau, D. D., Wu, N. Q., Hao, Q. Z., Pei, Y. P. Late Neogene evolution of the East Asian monsoon revealed by terrestrial mollusk record in Western Chinese Loess Plateau: from winter to summer dominated sub-regime, *Earth Planet. Sc. Lett.*, **274**, 439-447 (2008).

Liu, J., Song, X., Yuan, G., Sun, X., Yang, L. Stable isotopic compositions of precipitation in China, *Tellus B: Chemical and Physical Meteorology* **66**:1, 22567, doi:10.3402/tellusb.v66.22567 (2014).

Rousseau, D.D., Wu, N.Q. A new molluscan record of the monsoon variability over the past 130 000 yr in the Luochuan loess sequence, China. *Geology* **25**, 275–278 (1997).

Tang H., Micheels A., Eronen J., Fortelius M. Regional climate model experiments to investigate the Asian monsoon in the Late Miocene. *Clim Past* **7**(3), 847–868 (2011).

Wan S., Li, A., Clift, P. D., Stuu, J.-B. W. Development of the East Asian monsoon: Mineralogical and sedimentologic records in the northern South China Sea since 20 Ma. *Palaeogeogr. Palaeoclim. Palaeoecol.* **254**, 561-582 (2007).

Wu, N.Q., Rousseau, D.D., Liu, T.S., Lu, H.Y., Gu, Z.Y., Guo, Z.T., Jiang, W.Y. Orbital forcing of terrestrial mollusks and climatic changes from the Loess Plateau of China during the past 350 ka. *J. Geophys. Res.* **106**, 20045–20054 (2001).

A few minor points. In the Supplement- the wavelet spectral analyses appear look to me like the benthic $\delta^{18}\text{O}$ record has more consistent 100 kyr power than the planktonic,

which doesn't make particular sense if 41 kyr cycles dominate i.v. as the authors claim. More generally, it's not evident that the wavelet approach backs up the label on Fig 2 of main text for 7.2-7.7 Ma as "obliquity dominated".

Yes, the benthic $\delta^{18}\text{O}$ record has more consistent 100 kyr power than the planktonic record. However, this is massively dominated by the spectral power of obliquity in the benthic record, whereas the response to obliquity is weak in the planktonic record (see Blackman Tukey spectra included in Suppl. Fig. 3).

Suppl. Fig. 2 also indicates high benthic $\delta^{18}\text{O}$ wavelet spectral power in the 41 kyr obliquity band and unusually low spectral power in the 100 kyr eccentricity band for both $\delta^{18}\text{O}$ and orbits between 7.2 and 7.7 Ma (see Suppl. Fig. 2).

It's interesting that the standard deviation of Planktic $\delta^{18}\text{O}$ shows much more evolution over time than benthic.

We interpret this in terms of the intensity of the monsoon signal (Lines 181-196).

Please see specific comments annotated on the .pdf of the main text

We transferred these comments to this document, using line numbers of the submitted manuscript (see below).

Line 56-58: Isn't it circular to call cooling a response to temperature as a boundary condition

Agreed, we replaced temperature by ice volume (Line 56).

Line 59: The use of equilibrium may not be very helpful, since the early Pliocene saw a shift back to warmer conditions- the late Miocene changes do not represent irreversible ones

In the revised text, we avoided terminology referring to long-term system behavior; in particular, we now refrain from using the term "equilibrium state".

Line 103: I'm not sure that "transient" and "stepwise" are particularly helpful – they imply more analytical rigor than can be justified for what is in the end a visual analysis of such complex time series

We avoided these terms in the revised text

Lines 112-113: Since the changes occur in a context with strong orbital imprints, I'm not sure this language can be justified. There would appear to be more than one mode...

We revised our interpretation of benthic and planktic $\delta^{18}\text{O}$ trends (see Lines 97-141).

Lines 129-134: Are the authors concerned about the absolute values of SST inferred here relative to modern day?

We expanded the text dealing with SST reconstruction in Results (Lines 144-158) and in Supplementary Note 1 (Lines 112-139) to address this issue.

Line 156: see earlier comments re: attributing state change based purely on descriptive analysis

We thoroughly revised the text and avoided terms referring to system behavior and state changes (see responses to general comments above).

Line 159: explain why the SST change indicates a hydrological change

We now provide additional information, and references to back up our interpretation of hydrological change associated with intensification of the winter monsoon after 7 Ma (Lines 196-201 in Discussion; Lines 156-174 in Supplementary Note 1; Suppl. Fig. 10). Please also see detailed response to general comment above.

Line 172: Since the evolution of orbitally-related amplitudes, and in particular late Miocene glacial events occurred hundreds of kyr later, I don't see the idea of an abrupt change as illuminating

We completely revised the text here and abandoned the concept of a threshold response at 7.0 Ma in favor of a more evolutive concept of climate change, where the events around 7.2-7.0 Ma played an important role in initiating a longer-term cooling trend (also see answer to general comment above).

Lines 192 to 199: whether this is ultimately right or wrong, it is good example of creative, important interpretation of a new result

The hypothesis of high-latitude controls of low latitude biological productivity and its sensitivity to climate change through intermediate water nutrient transfer (Sarmiento et al., 2004, Sigman et al., 2010) is probably important for understanding feedbacks not only in glacial/interglacial changes and future global warming but also for late Miocene climate oscillations.

Sarmiento, J.L., Gruber, N., Brzezinski, M.A., Dunne, J.P. et al. High-latitude controls of thermocline nutrients and low latitude biological productivity. *Nature* 427, 56-60 (2004).

Sigman, D.M., Hain, M.P. & Haug, G.H. The polar ocean and glacial cycles in atmospheric CO₂ concentration. *Nature* 466, 47-55. doi:10.1038/nature09149 (2010).

Lines 209-211: we would expect a warm pool location to have a high pump efficiency so I'm not sure this is justified. In other words, how likely is it that the SCS is a place where we would see this kind of change?

We agree with the reviewer that the term “efficiency of the biological pump” here is ambiguous. The efficiency of the biological pump is commonly defined as the amount of carbon exported from the surface layer divided by the total amount produced through photosynthesis (e.g. Ducklow et al., 2001). Strengthened monsoonal winds would increase the new production and consequently the carbon export flux, which is dependent on the new production, but would not necessarily increase the efficiency of the biological pump. In view of this, we replaced “more efficient biological pump” by “increased sequestration efficiency” or “increased carbon sequestration flux” (Lines 243, 252-253, 372-373).

Fig. 9 a Comparison of modeled $p\text{CO}_2$ with observations at the SEATS station (Sheu et al. 2010). b Modeled monthly mean sea-to-air CO_2 flux over the entire basin. Atmospheric $p\text{CO}_2$ close to SEATS (Sheu et al. 2010) is used to calculate the sea-to-air CO_2 flux of the basin

from Ma et al. (2014)

Present day observations and model simulations of the biological pump structure in the SCS (Ma et al., 2014) indicate significant differences in sea-to air CO₂ fluxes depending on the monsoonal season. Positive fluxes characterize the summer monsoon conditions, negative fluxes the winter conditions. We assume that similar changes may have occurred on orbital time scales in the Late Miocene.

Even fluxes into the sedimentary reservoir (burial flux) may have played a role on the long timescales we are considering here. Burial flux does not contribute much to maintain the gradient of carbon concentration between the mixed layer and the interior of the ocean on millennial timescales, but it may contribute to significant changes in the carbon budget between the ocean/sedimentary and atmospheric reservoirs on orbital (in particular eccentricity) and longer timescales.

Ducklow, H.W., Steinberg, D.K. & Buesseler, K.O. Upper Ocean Carbon Export and the Biological Pump. *Oceanography*, 14/4 (Special Issue JGOFS), 50-58 (2001).

Ma, W., Chai, F., Xiu, P., Xue, H. and Tian, J. Simulation of export production and biological pump structure in the South China Sea. *Geo-Marine Letters* 34, 541-554, doi 10.1007/s00367-014-0384-0 (2014).

Lines 215-216:... and a general rule is that high export production actually equates with lower pump efficiency

This issue is solved with the changes implemented in Lines Lines 243, 252-253, 372-373.

Lines 228-229 (re-organization in SE Asian subtropical climate at ~7 Ma marked a significant step in late Miocene climate evolution) – see comments on similar claims earlier – other proxies of SE Asian climate besides marine stable isotopes?

We no longer argue for a fundamental change in the system behavior at ~7 Ma. We now refer to other studies supporting a major hydrological change after 7 Ma (see previous answer to general comment above).

Lines 233-239: While I like the effort to get to climatic interpretation, the mean d18O values from 6.5-7Ma are about the same as 7.5-9Ma

See response to general comment above.

Lines 251-252 cite literature in support of this?

In the text, we now refer to 2 studies that provide evidence for episodes of Northern Hemisphere ice buildup between 6 and 5 Ma (Larsen et al., 1994; Krissek, 1995). Additional references for late Miocene NH glaciations include Thiede et al. (1998) and St. John et al. (2002). Several late Miocene IRD pulses at ODP Site 909 (Fram Strait) indicate stepwise increase of NH cooling between 10.8 and 8.6 Ma, around 7.2, 6.8 and 6.3 Ma (Thiede et al., 1998). In the late Miocene (~7.3 Ma), IRD supply to Ocean Drilling Program site 918 increased significantly indicating that glaciers large enough to reach

sea level were present in SE Greenland long before the onset of widespread Northern Hemisphere glaciation (St. John et al., 2002).

Krissek, L. Late Cenozoic ice-rafting records from Leg 145 sites in the North Pacific: Late Miocene onset, Late Pliocene intensification, and Pliocene-Pleistocene events. In *Proceeding of the Ocean Drilling Program, Scientific Results 145* (eds Rea, D. K., Basov, I. A., Scholl, D. W., Allan, J. F.) 179-194 (Ocean Drilling Program, 1992).

Larsen, H. C., Saunders, A. D., Clift, P. D., Beget, J., Wei, W., Spezzaferri, S. Seven million years of glaciation in Greenland. *Science* **264**, 952-955 (1994).

St. John, K. E. K. & Krissek, L. A. The late Miocene to Pleistocene ice-rafting history of south-east Greenland. *Boreas*, 31, 28–35. Oslo. ISSN 0300-9483 (2002).

Thiede, J., Winkler, A., Wolf-Welling, T., Eldholm, O., Myhre, A. M., Baumann, K.-H., Henrich, R., Stein, R. Late Cenozoic history of the polar North Atlantic: Results from Ocean Drilling. *Quaternary Science Reviews*, 17, 185-208 (1998).

Lines 261-262: If moisture transport is really what's key ... why not simpler claim of less summer insolation at high latitudes when obliquity is low?

According to Raymo and Nisancioglu (2003) and Loutre et al. (2004), meridional moisture transport during periods of high differential heating between high and low latitudes at obliquity minima would promote ice-sheet growth, whereas a low summer insolation gradient at obliquity maxima would decrease poleward moisture transport, inhibiting ice-sheet build-up. Obliquity-paced variations in moisture supply for the late Pleistocene East Antarctic ice-sheet are also evident from the deuterium excess record in the Vostok ice core (Vimeux et al., 2001). The underlying assumption is that the dynamics of the late Miocene Antarctic ice sheet was similar to that of the Holocene. The controlling factor of ice sheet growth is the lack of precipitation and not ablation by summer insolation (summers in central Antarctica are still cold enough to prohibit significant melting of the ice sheet). Growth of the ice sheet, thus, will more likely be promoted by supply of more snow precipitation rather than during periods of cold/dry conditions. To clarify this point, we added “and lower summer insolation at high latitudes” in the main text (Line 301).

Raymo, M. E. & Nisancioglu, K. The 41 kyr world: Milankovitch's other unsolved mystery. *Paleoceanography* 18, 1011 doi:10.1029/2002PA00791 (2003).

Loutre, M.-F., Paillard, D., Vimeux, F. & Cortijo, E. Does mean annual insolation have the potential to change the climate? *Earth Planet. Sci. Lett.* 221, 1-14 ((2004).

Vimeux, F. et al. A 420,000 year deuterium excess record from East Antarctica: Information on past changes in the origin of precipitation at Vostok. *J. Geophys. Res.* 106, 31863–31873 (2001).

Line 284: the period does NOT appear warmer if you look to the planktonic d18O.

We refer here to the middle Miocene, in particular to the transition from the Climatic Optimum, which was warmer than the late Miocene interval, to a cooler climate mode following expansion of the Antarctic ice sheet.

Line 290-291: I would strike this – again it is hard to separate orbital-scale changes from state changes

Agreed, we deleted the sentence.

Line 295-296 why invoking sustained feedbacks and/or CO₂ forcing, this doesn't make sense, since radiative forcing from orbital variations does not have direction over time

We agree that radiative forcing from orbital variations does not have a direction over very long time scales (>1.2 Ma). However, it does have direction over periods of up to 1.2 Ma (very long obliquity and eccentricity cycles as illustrated in Suppl. Fig. 6). These trends may be influential in driving the climate system over critical thresholds.

Lines 297-298: could you back up the claim on monsoon better?

See response to general comment above.

Line 301-303: while I agree in principle, there is no actual demonstration of threshold behavior as it is inferred rather than demonstrated (e.g. CO₂ proxies are inadequate to back up the claim at this point)

We revised our interpretation and abandoned the concept of a threshold response at 7.0 Ma in favor of a more evolutive concept of climate change (see revised title, Lines 20-23 in abstract, Discussion, Lines 329-342 in final paragraph).

Reviewer #3 (Remarks to the Author):

Holbourn et al. have constructed orbital-scale stable isotope and Mg/Ca records from ODP Site 1146, South China Sea, spanning the late Miocene (9-5 Ma). The focus of the study is a warming event between 7.7-7.2 Ma followed by a cooling event 7.2-7.0 Ma, which is to mark a major climate re-organization (e.g., line 295). The data are of high quality, and generally the manuscript is well written.

What the authors refer to as a ‘major’ cooling event is exhibited as a 0.3 per mil increase in benthic foram $\delta^{18}\text{O}$ values (~ 1 degree C) and a 2 degree C cooling of mixed layer temperatures. Thus in comparison to major events such as at the Eocene/Oligocene boundary, the mid Miocene, or the onset of significant Northern Hemisphere Glaciation, this one seems relatively minor. But this does not mean that it is not worth looking at in greater detail.

During revision, we toned down the importance of the 7.2-7.0 Ma climate change as a single, unique event. Instead, we consider that this transition plays an important role in initiating a long-term cooling trend culminating with transient Northern Hemisphere glaciations (TG events) before the climate reversal in the early Pliocene (see revised title, Lines 20-23 in abstract, Lines 269-276 in Discussion, Lines 329-342 in final paragraph).

The authors argue for ‘substantially higher amplitude variability’ (line 106) of the climate signal after the cooling event at 7.2-7.0 Ma. I suggest to show this more quantitatively using time series analysis. At the very least, they should mention whether or not the difference in the variability is statistically significant (lines 106/107).

The increase in amplitude variability is shown in the moving window standard deviation plot in Suppl. Fig. 6. We now refer to this figure in the text (Lines 121-125). We also provide additional time series and statistical analyses (Blackman-Tukey Power Spectra, coherence and phase analyses, locally weighted least squared error, moving window correlation; see Suppl. Figs. 3-4, 14-15) that allow a more rigorous quantitative evaluation of data and interpretations. We refer to these additional quantitative results in relevant parts of the main text (see in particular Lines 97-172 in Results, 270-276 in Discussion) and in the Supplement text.

Wavelet power spectra of planktonic $\delta^{18}\text{O}$ indicate a shift of spectral power towards longer wavelengths after 7.2 Ma (Suppl. Fig. 2F). The comparison of $\delta^{18}\text{O}_{\text{planktonic}}$ and $\delta^{18}\text{O}_{\text{benthic}}$ Blackman-Tukey power spectra for the intervals 5-7.15 Ma and 7.15-9 Ma indicate a generally higher response to orbital forcing between 7.15 and 9 Ma with a dominant response of benthic $\delta^{18}\text{O}$ to 41 kyr obliquity and a stronger response of planktic $\delta^{18}\text{O}$ to 23 kyr precession (Suppl. Fig. 3)

I don't think that planktonic-benthic foraminiferal $\delta^{13}\text{C}$ gradients are a commonly accepted proxy for atmospheric CO_2 levels (Figure 3A) (discussion beginning on line 200). Although the authors make an argument about how it works conceptually, I think in reality the interpretation of the gradient is more complicated than stated. For example, theoretically, the gradient is also reduced when benthic foraminiferal $\delta^{13}\text{C}$ values increase, which can be due to processes in the source region of deep waters and variations in the relative proportions of deep water masses. How does a planktic-benthic $\Delta\delta^{13}\text{C}$ record for the past 800,000 years compare with the ice core pCO_2 record? I think the text needs a more thorough discussion of all the factors that can affect the vertical carbon isotope gradient as well as proof of concept (supplemental maybe?).

We followed this suggestion and plotted the late Pleistocene Dd13C record from ODP Site 1146 against the measured pCO₂ from the EPICA ice core. Taking into account the independent age models, Dd13C does reflect pCO₂ variability, in particular on a glacial-interglacial scale. We added this comparison in Suppl. Fig. 12 and refer to it in the Discussion (Lines 252-254) and Supplementary Note 2 (Lines 194-198).

We agree that the relationship of Dd13C with pCO₂ is complex and added a more detailed discussion of factors influencing the Dd13C record at Site 1146 in Supplementary Note 2.

There are several sentences throughout the text that seem to make strong statements (e.g., line 222 and line 262, see below), but the underlying evidence, or at the very least citations, are not provided.

Steepening of equator to pole gradient after 7 Ma is evident from the compilation of late Miocene temperature records in Herbert et al. (2016). We now include a comparison of these records with the 1146 subtropical mixed layer temperature reconstructions in Suppl. Fig. 9 and we refer to this new figure on Lines 262-265 in Discussion.

According to Raymo and Nisancioglu (2003) and Loutre et al. (2004), meridional moisture transport during periods of high differential heating between high and low latitudes at obliquity minima would promote ice-sheet growth, whereas a low summer insolation gradient at obliquity maxima would decrease poleward moisture transport, inhibiting ice-sheet build-up. Obliquity-paced variations in moisture supply for the late Pleistocene East Antarctic ice-sheet are also evident from the deuterium excess record in the Vostok ice core (Vimeux et al., 2001). The underlying assumption is that the dynamics of the late Miocene Antarctic ice sheet was similar to that of the Holocene. The controlling factor of ice sheet growth is the lack of precipitation and not ablation by summer insolation (summers in central Antarctica are still cold enough to prohibit significant melting of the ice sheet). Growth of the ice sheet, thus, will more likely be promoted by supply of more snow precipitation rather than during periods of cold/dry conditions. To clarify this point, we added “and lower summer insolation at high latitudes” in the main text (Line 301).

Raymo, M. E. & Nisancioglu, K. The 41 kyr world: Milankovitch's other unsolved mystery. *Paleoceanography* 18, 1011 doi:10.1029/2002PA00791 (2003).

Loutre, M.-F., Paillard, D., Vimeux, F. & Cortijo, E. Does mean annual insolation have the potential to change the climate? *Earth Planet. Sci. Lett.* 221, 1–14 (2004).

Vimeux, F. et al. A 420,000 year deuterium excess record from East Antarctica: Information on past changes in the origin of precipitation at Vostok. *J. Geophys. Res.* 106, 31863–31873 (2001).

Figure 4 is only mentioned in passing on lines 137 and 149. I think it shows a lot of information and needs a more thorough description with reference to the specific panels. Details of the climate events between 7.4 and 6.9 Ma are now described on Lines 110-

114, 122-125, 177-185 (changes in oxygen isotopes and mixed layer temperatures) and Lines 137-141 (increase in planktonic-benthic d13C gradient).

In order:

Line 56-59: The sentence is awkward. In essence it means that cooling can be a response to temperature...

Agreed, we replaced temperature by ice volume (Line 56).

Line 222-225: What is the evidence for this statement? I think this needs citations to specific articles.

Evidence for biogenic blooms in the Pacific are provided in Lyle and Baldauf (2015 and references therein). We refer to this study on Lines 206, 261, 267.

Line 236 and again line 246: The author use terms that imply statistical analyses (covariance, in phase..), but none are provided, at least not in this context?

We now provide the results of the cross spectral analysis between planktonic and benthic d18O in Suppl. Fig. 3 and refer to this figure on Lines 97-102 in Results.

Line 260: 'minima in obliquity': I think they mean minima in the amplitude of obliquity variations (a node)?

We mean obliquity minima, not nodes (minima in amplitude variability).

Line 261-263: How exactly does this work? I think the idea that glaciations occur during times of obliquity nodes has been discussed in the literature (Hilgren's work?).

Please refer to detailed response to general comment above with relevant references. To clarify this point, we added "and lower summer insolation at high latitudes" in the main text (Line 301).

Line 279: Given Cenozoic ice-volume history, I am not sure that a 0.2- to 0.3 per mil increase in d18O indicates 'substantial' growth of ice and or deep water cooling. I would suggest eliminating qualifying words like 'major', 'substantial' or 'striking'.

We followed this suggestion and eliminated "substantial" and similar qualifying terms.

Reviewers' comments:

Reviewer #1 (Remarks to the Author):

This is a re-review of "Late Miocene global cooling linked to subtropical climate re-organization", by Holbourn et al. I am reviewer 1, from the previous review. It is a good sign to me that all three of the reviewers had overlapping and complementary reviews since this at least lays out clearly the nature of the desired changes. My re-review will be short because, while I see that the authors have taken a first step toward the revisions I suggested, very little in the main manuscript has actually changed and the problems identified remain.

Many claims are still made without a proper backing or even a definition of what is meant. For example, what exactly is a "subtropical climate re-organization"? The authors now include some more time series in supplemental figure 9, but those show a general cooling. In all that ways that are interesting, concomitant cooling (within resolution) at a variety of sites is not a reorganization. Atmosphere and ocean circulations are driven by gradients in temperature (density), so reorganizations are associated with changes in spatial gradients. If there is a point to be made here (let alone prominently displayed in the title) the authors will need to make maps of temperature/density in specific time slices and the relate them to other parameters (such as monsoon precipitation/freshwater records).

Very little has changed in this submission unfortunately. Consider the captions written onto Figure 2,3,4 ("re-organization of global carbon reservoir", "decreasing pco2") and then a bunch of arrows. We now have quantitative frameworks for evaluating such statements, they should be used.

Similarly a big deal is still being made about an event at 7.2 Ma, but in the response to reviewers (and to a lesser degree in the text) it is characterized now as a 'mode' or regime shift. This is still very hand waving. What regime? What mode? How are these quantitatively defined? What independent lines of evidence or modeling can be brought in to characterize this new mode? Simply citing the fact that other papers have identified changes during the same interval actually undermines the novelty of this paper.

The authors need to make it clear what we understand now that we did not before?

Reviewer #2 (Remarks to the Author):

Before I began my review, I re-read all three reviewer's comments, which shared a lot in common. We all agreed that the data are of extremely high quality and that the text should provide an important contribution to a period of time that is yielding paleoclimatic surprises. After then reading the author's replies, the revised manuscript, and the revised supplement, I feel that they have done a very thorough job in responding constructively to critical parts of the reviews.

In particular:

- The tone of the analysis shifts in general to a more evolutive view of the climate change captured in the proxy data sets.
- The text and supplement spend more attention to backing up claims with statistics and spectral analyses
- The text provides a more nuanced interpretation of Dd13C gradients as a proxy for ocean CO2 storage.

There are still aspects I would disagree with, particularly the author's assumption that the timing of the first surface cooling in their record is a global phenomenon, but I think that the authors have gained the right to make their case. I therefore recommend publication, with some attention to the following points.

Comment

Lines 261-262: If moisture transport is really what's key ... why not simpler claim of less summer insolation at high latitudes when obliquity is low?

"According to Raymo and Nisancioglu (2003) and Loutre et al. (2004), meridional moisture transport during periods of high differential heating between high and low latitudes at obliquity minima would promote ice-sheet growth, whereas a low summer insolation gradient at obliquity maxima would decrease poleward moisture transport, inhibiting ice-sheet build-up. Obliquity-paced variations in moisture supply for the late Pleistocene East Antarctic ice-sheet are also evident from the deuterium excess record in the Vostok ice core (Vimeux et al., 2001). The underlying assumption is that the dynamics of the late Miocene Antarctic ice sheet was similar to that of the Holocene. "

In the late Pleistocene, the East Antarctic ice sheet has almost no room to grow. It is not clear that supposing that Miocene dynamics would be similar to the late Pleistocene (= "Holocene" here?) is a good idea. I would also emphasize that the Raymo and Loutre papers present results from MODELS- it is far from clear that the models capture reality. And while deuterium excess records may reflect obliquity-paced variations in moisture supply, they do not constrain the net accumulation or ablation of the East Antarctic ice sheet. Therefore, I find the unique emphasis on moisture supply to go beyond what can be substantiated.

Line 53: "However, the precise onset and trigger of this extended cooling episode " seems a hold-over from the previous text. Why does an "onset" need to be necessarily "precise"? This

assumes that the climate transition was necessarily rapid, hence, precisely identifiable, which flies in the face of the author's revisions elsewhere to emphasize a more evolutive point of view.

Furthermore, I can assure the authors that the cooling they observe at 7.2 Ma at Site 1146 may not be representative of a particularly well-defined GLOBAL surface cooling- there are enough paleotemperature records through this time to establish that in many, if not most places, cooling had begun earlier. These studies would confirm that cooling accelerated after ~7.2 Ma.

. Line 178-179: "our data reveal that the high-latitude cooling trend, which started after 7.2 Ma, coincided with a re-organization in subtropical climate

. ". The authors have NOT established a high latitude cooling trend- they are looking at benthic $\delta^{18}O$ which may or may not reflect generalized high latitude temperature changes. High latitude sites show significant cooling prior to 7.2 Ma in the Herbert et al. (2016) compilation. While none of those sites got to true polar locations, they do represent "high latitudes" in the general sense.

. I'm having trouble with the interpretation of changes in interbasin $\delta^{13}C$ gradients- in particular the inference of preformed $\delta^{13}C$ values

. Lines 229-230: Thus, our data support that changes in preformed $\delta^{13}C$ and increased export of water with a higher preformed nutrient composition to the lower latitudes drove the sharper interbasinal $\delta^{13}C$ gradient. " Since there are multiple sources of deep water formation, to which does this interpretation apply?

. Lines 238-241: "The convergence of the $\delta^{13}C$ records after 7.2 Ma suggests

expansion of a $\delta^{13}C$ depleted central Pacific deep water mass into shallower depths. The expansion of a ^{12}C -enriched deep water mass would also imply increased carbon storage in the deep Pacific Ocean after 7.2 Ma. "

What's confusing is that in general, high preformed nutrient content of bottom water implies an inefficient biological pump and low values of sequestered CO_2 in the deep ocean.

Lines 253-255: This effect additionally becomes accentuated with an increased carbon sequestration flux, for instance, during periods when monsoonal winds strengthen, as shown in the Site 1146 late Pleistocene record (Suppl. Fig. 12).

This would imply that local carbon flux plays an important role in the global storage of carbon in the deep ocean- an idea that is undercut elsewhere in the discussion that emphasizes physical and biochemical aspects of deep water formation as the primary controls on deep ocean carbon sequestration. Likewise, equatorial Pacific productivity generally is neglected in global ocean

carbon cycle models which instead emphasize productivity and nutrient utilization in areas of deep and intermediate water formation.

. Lines 306-309: “Renewed 100 kyr eccentricity forcing after 7.2 Ma, combined with high-amplitude variations in obliquity, probably drove the successive rebounds between 7.2 and 7.0 Ma, but did not succeed in reversing the cooling trend. “

I would avoid using the term “forcing” with “eccentricity” because the insolation variations driven by eccentricity are so small that it is very hard to imagine they directly force the climate. Instead, the dominant view is that non-linearities in climate feedbacks to precessional variations end up producing an output with the eccentricity modulation imprint.

. Lines 330-333: “In particular, our results show that ice volume was not the primary driver of global climate and that low latitude processes including monsoonal winds, large-scale ocean circulation and productivity played a fundamental role in setting late Miocene climate trends. “

While one can agree that the present study uncovers very interesting variability in proxies related to low latitude processes, it hasn't demonstrated that these “set” late Miocene climate trends. In fact, subsequent text proposes the global carbon cycle as the driver.

Reviewer #3 (Remarks to the Author):

There is no doubt that the manuscript by Holbourn et al presents high quality geochemical data sets that reflect late Miocene climate evolution in unprecedented detail. As a result, the manuscript contains 23 figures, eight in the main text and 15 in the supplemental information. In order to make such an enormous amount of information readable, results need to be communicated effectively by following a consistent structure. However, this manuscript is difficult to follow at times. Many times I am not sure whether or not I am correctly understanding what the authors are trying to say. So I am unsure whether or not my comments regarding the scientific content are valid and pertinent. My review provides some examples of ambiguities, and I think the authors should revise all of the manuscript with an eye for clarity.

Figure call-outs: One problem with the manuscript organization is that the text refers to specific information presented in figures, but then calls out to a range of figures without specifying in which panel the information can be found. Conversely, figures contain panels that are not addressed in the same context as other panels of the same figure. I feel like the reader is being asked to mentally fill in these gaps, which is time consuming.

An example of confusing paragraph structure:

The topic sentence on line 97 points to differences in long-term trends between benthic and planktonic d18O records, as well as coherence between them at the obliquity-scale. Then the authors simply call out to Figure 2 as well as Supplemental Figures 3-4 (line 98). The reader has to search Figure 2 for the correct panels (B and C), and is also left to search for the specific differences in the long-term trends between these two records. The text does not provide any supporting statements about how the long-term trends shown in Figure 2 are different. Supplemental Figure 3 presents the results from the spectral analysis. However, it only shows power spectra, the text on the other hand, refers to “high coherence” (line 98). The coherence between benthic and planktonic d18O is not shown in this figure (despite their response in the rebuttal to be showing this). Supplemental Figure 4 shows the time series of both records in panels A and B with long-term trends superimposed together with a comparison of the d18O records to corresponding d13C records.

In my mind, Supplemental Figure 4 presents the information that fits with the main text of this paragraph (beginning on line 97). Panels A and B very nicely illustrate similarities and differences in the planktic and benthic long-term d18O trends (they are illustrated by a smoothed curve). Then, if the topic sentence of this paragraph is to stand as is (first mentioning the long-term trend and then the short-term trend), it should be followed first by the supporting points regarding the long-term trends second the supporting points regarding the short-term trends. The orbital-scale variations should be addressed calling out to a figure that shows the cross correlation. And, in order to fit the d13C into this paragraph, the topic sentence could be modified to read: "Benthic and planktic stable isotope records exhibit..." In this manner the pertaining d18O and d13C trends can be fully described before addressing the orbital-scale variations.

Line 107: It is not clear to me what the standard deviations refer to. Are they standard deviations of averages over discrete time intervals?

Line 128: Sentence and call out to Supplemental Figure 7: Is it possible that the color scale of the coherence is wrong? As is, it looks like the background color yellow is also the color that defines highest coherence? It would suggest that the records are highly coherent everywhere except in the well-defined blue areas? The blue, circled areas are those of lowest coherence? The captions does point out, however, that the circled regions (which are blue) are 5% significance against red noise.

The manuscript text states that at the 41 kyr periodicity, the planktic and benthic d13C values are coherent throughout the time interval. But in the Supplemental Figure 7, the blue regions are far less prominent compared to the yellow background region in the roughly 30-64 kyr band. However, in the precession band, the blue regions dominate. So, is this a typo on line 129?

Lines 179-187: This part of the discussion read more like results. It provides detail that should be described in the context of the description of the time series.

Lines 269-272: Figure 7 shows the entire benthic d13C and d18O records from Site 1146. How do these records illustrate "re-organization of the SE Asian monsoon"? And, why is there a label "D" in the lower right hand corner?

Lines 283-284: "in-phase" relationship between benthic d18O maxima and planktic d18O increases: This sentences is confusing. First, this statement implies that cross spectral analyses were conducted, the authors call out to Figure 5, which shows time series. Second, I am not sure what the authors mean. As written, the sentence implies that planktic d18O values still increase while benthic d18O values have already reached a maximum. In a cross spectral analysis such a relationship would show up as a lead/lag, should it be statistically significant. On the other hand, the same sentence also says that the records indicate synchronous variations between deep water d18O and NH climate. But looking at the planktic and benthic d18O records, the two are not always synchronous. Some of the maxima occur at the same time in both records (TG22 and TG 20), but others are not (TG4). I think the wording needs to be more precise.

Response to Reviewers' comments:

Reviewer #1 (Remarks to the Author):

This is a re-review of "Late Miocene global cooling linked to subtropical climate re-organization", by Holbourn et al. I am reviewer 1, from the previous review. It is a good sign to me that all three of the reviewers had overlapping and complementary reviews since this at least lays out clearly the nature of the desired changes. My re-review will be short because, while I see that the authors have taken a first step toward the revisions I suggested, very little in the main manuscript has actually changed and the problems identified remain.

Many claims are still made without a proper backing or even a definition of what is meant. For example, what exactly is a "subtropical climate re-organization"? The authors now include some more time series in supplemental figure 9, but those show a general cooling. In all that ways that are interesting, concomitant cooling (within resolution) at a variety of sites is not a reorganization. Atmosphere and ocean circulations are driven by gradients in temperature (density), so reorganizations are associated with changes in spatial gradients. If there is a point to be made here (let alone prominently displayed in the title) the authors will need to make maps of temperature/density in specific time slices and the relate them to other parameters (such as monsoon precipitation/freshwater records).

Very little has changed in this submission unfortunately. Consider the captions written onto Figure 2,3,4 ("re-organization of global carbon reservoir", "decreasing pCO₂") and then a bunch of arrows. We now have quantitative frameworks for evaluating such statements, they should be used.

The manuscript was substantially revised to address constructive comments from reviewers. In response to criticisms from Reviewer 1, we re-organized the interpretation and discussion sections, focusing on three key findings derived from the new, highly resolved proxy time series presented in this paper:

1. Long-term climate cooling from ~7 Ma until ~5.5 Ma was synchronous with intensification of the Asian winter monsoon.
2. Steepening of the gradient between planktic and benthic $\delta^{13}\text{C}$ after ~7 Ma, coincident with increased productivity and nutrient regeneration in the Pacific Ocean ("biogenic bloom"), indicates pCO₂ decrease and increased carbon storage in the deep ocean.
3. Climate cooling after ~7 Ma culminated with ephemeral Northern Hemisphere glaciations (TG cold events) between 6.1 and 5.5 Ma.

In our revision, we avoided loosely-defined terms such as “subtropical climate re-organization” and clarified text and figures, deleting speculative passages in the discussion and over-interpretative arrows and labels in the main figures (such as “re-organization of global carbon reservoirs” in Fig. 2). We nevertheless retained the label “decreasing pCO₂” in Figs. 2-5, as this interpretation is backed up by steepening of the gradient between planktic and benthic $\delta^{13}\text{C}$ (as shown in Fig. 3 and discussed on Lines 246-273 and Supplementary Note 4). The title was changed to “Late Miocene climate cooling and intensification of SE Asian winter monsoon” to highlight a main finding of the paper and to avoid speculative over-interpretation.

However, the suggestion to include more quantitative frameworks such as maps of temperature/density in specific time slices to better evaluate and interpret data is, in our opinion, unrealistic. Firstly, the chronology of late Miocene records is still extremely poorly constrained and, in most cases, does not resolve orbital-scale climate variability, thus precluding detailed correlations of short-term and orbital-scale climate events. Second, current pCO₂ reconstructions over the studied time interval are extremely sparse and have very large error bars (e.g., Zhang et al., 2013; Herbert et al., 2016); thus, quantitative estimates of atmospheric pCO₂ remain highly speculative. Third, we think that including a modeling component would not significantly advance data interpretation and that it is beyond the scope of the current ms and should be the target of an independent study. As pointed out by Reviewer 2, it is far from clear that current models capture reality, especially for intervals such as the late Miocene, when boundary conditions including paleogeography (in particular, gateway configuration and Himalayan tectonics), atmospheric pCO₂, ice volume and ocean circulation are still extremely poorly understood.

In our opinion, the strength of our paper lies in providing a new, unique set of highly resolved and chronologically well-constrained proxy data that allow unprecedented insights into the timing of late Miocene climate changes and into the evolution of upper and deep ocean water mass properties, which will also help to define more reliable targets for modeling studies.

Similarly a big deal is still being made about an event at 7.2 Ma, but in the response to reviewers (and to a lesser degree in the text) it is characterized now as a 'mode' or regime shift. This is still very hand waving. What regime? What mode? How are these quantitatively defined? What independent lines of evidence or modeling can be brought in to characterize this new mode? Simply citing the fact that other papers have identified changes during the same interval actually undermines the novelty of this paper.

We no longer use these loosely-defined terms (see previous response).

In our opinion, referring to independent lines of evidence such as paleobotanic studies that support the establishment of drier, cooler conditions and intensified

winter monsoon on the Asian land mass, does not undermine the novelty of this paper. On the contrary, establishing a connection between land and marine records is essential to demonstrate that changes are not restricted to one region, but have a supra-regional or global impact at low and intermediate latitudes.

Second, benthic and planktic oxygen and carbon isotopes still provide the most robust climate proxy data beyond the Quaternary, allowing tracking of hydrosphere variability (temperature/salinity, ice volume, circulation and deep/intermediate water ventilation) at unprecedented resolution as well as testing of less conventional or more ambiguous proxy data.

Third, benthic oxygen isotopes provide the backbone of the Cenozoic chronology and are crucial to anchor and validate more loosely-dated reconstructions. For instance, synthesis of regional and benthic isotope data provided a better understanding of Miocene cryosphere-climate evolution and ice-sheet dynamics (e.g., Levy et al., 2016, Fig. 2 and Suppl. Fig. S5 at www.pnas.org/cgi/doi/10.1073/pnas.1516030113).

The authors need to make it clear what we understand now that we did not before?

The major steps forward in understanding Miocene climate dynamics and long-term evolution are:

1. Climate development on the Asian landmass was largely decoupled from the evolution of the Antarctic ice sheet, which formed the main component of the cryosphere during the late Miocene.
2. Late Miocene climate cooling after ~7 Ma was synchronous with increased productivity and nutrient regeneration, potentially driving increased carbon storage in the deep ocean and pCO₂ decrease.
3. Upper ocean hydrological change indicates intensification of SE Asian winter monsoon between 7 and 5.5 Ma.
4. Climate cooling after ~7 Ma culminated with ephemeral Northern Hemisphere glaciations (TG cold events) between 6.1 and 5.5 Ma.
5. Carbon cycle (benthic and planktic d13C variations) show strong response to insolation forcing at all main orbital bands.
6. Improved chronology for late Miocene climate events.

Reviewer #2 (Remarks to the Author):

Before I began my review, I re-read all three reviewer's comments, which shared a lot in common. We all agreed that the data are of extremely high quality and that the text should provide an important contribution to a period of time that is yielding paleoclimatic surprises. After then reading the author's replies, the revised manuscript, and the revised supplement, I feel that they have done a very thorough job in responding constructively to critical parts of the reviews.

In particular:

- The tone of the analysis shifts in general to a more evolutive view of the climate change captured in the proxy data sets.
- The text and supplement spend more attention to backing up claims with statistics and spectral analyses
- The text provides a more nuanced interpretation of $\delta^{13}C$ gradients as a proxy for ocean CO₂ storage.

There are still aspects I would disagree with, particularly the author's assumption that the timing of the first surface cooling in their record is a global phenomenon, but I think that the authors have gained the right to make their case. I therefore recommend publication, with some attention to the following points.

Comment

Lines 261-262: If moisture transport is really what's key ... why not simpler claim of less summer insolation at high latitudes when obliquity is low?

"According to Raymo and Nisancioglu (2003) and Loutre et al. (2004), meridional moisture transport during periods of high differential heating between high and low latitudes at obliquity minima would promote ice-sheet growth, whereas a low summer insolation gradient at obliquity maxima would decrease poleward moisture transport, inhibiting ice-sheet build-up. Obliquity-paced variations in moisture supply for the late Pleistocene East Antarctic ice-sheet are also evident from the deuterium excess record in the Vostok ice core (Vimeux et al., 2001). The underlying assumption is that the dynamics of the late Miocene Antarctic ice sheet was similar to that of the Holocene. "

In the late Pleistocene, the East Antarctic ice sheet has almost no room to grow. It is not clear that supposing that Miocene dynamics would be similar to the late Pleistocene (= "Holocene" here?) is a good idea. I would also emphasize that the Raymo and Loutre papers present results from MODELS- it is far from clear that the models capture reality. And while deuterium excess records may reflect obliquity-paced variations in moisture supply, they do not constrain the net accumulation or ablation

of the East Antarctic ice sheet. Therefore, I find the unique emphasis on moisture supply to go beyond what can be substantiated. We have modified the text to emphasize both aspects of minima in obliquity or minima in the amplitude variability of obliquity (“nodes”): increased gradient between low and high latitude insolation that drives latitudinal heat and moisture transport and reduced high latitude summer insolation

Line 53: “However, the precise onset and trigger of this extended cooling episode “ seems a hold-over from the previous text. Why does an “onset” need to be necessarily “precise”? This assumes that the climate transition was necessarily rapid, hence, precisely identifiable, which flies in the face of the author’s revisions elsewhere to emphasize a more evolutive point of view.

In response to this criticism, we re-formulated the last two sentences of the paragraph. In particular, we no longer refer to the onset of the climate transition (Lines 60-63).

Furthermore, I can assure the authors that the cooling they observe at 7.2 Ma at Site 1146 may not be representative of a particularly well-defined GLOBAL surface cooling- there are enough paleotemperature records through this time to establish that in many, if not most places, cooling had begun earlier. These studies would confirm that cooling accelerated after ~7.2 Ma.

We agree that cooling may not be globally synchronous. However, synchronicity is difficult to assess due to the different resolution of data sets and potential incompatibility of age models.

. Line 178-179: “our data reveal that the high-latitude cooling trend, which started after 7.2 Ma, coincided with a re-organization in subtropical climate”. The authors have NOT established a high latitude cooling trend-they are looking at benthic d18O which may or may not reflect generalized high latitude temperature changes. High latitude sites show significant cooling prior to 7.2 Ma in the Herbert et al. (2016) compilation. While none of those sites got to true polar locations, they do represent “high latitudes” in the general sense.

Sentence was deleted.

I’m having trouble with the interpretation of changes in interbasin d13C gradients- in particular the inference of preformed d13C values
Lines 229-230: Thus, our data support that changes in preformed d13C and increased export of water with a higher preformed nutrient composition to the lower latitudes drove the sharper interbasinal d13C gradient. “ Since there are multiple sources of deep water formation, to which does this interpretation apply?

See response below.

Lines 238-241: “The convergence of the $\delta^{13}\text{C}$ records after 7.2 Ma suggests expansion of a $\delta^{13}\text{C}$ depleted central Pacific deep water mass into shallower depths. The expansion of a ^{12}C -enriched deep water mass would also imply increased carbon storage in the deep Pacific Ocean after 7.2 Ma.”

What’s confusing is that in general, high preformed nutrient content of bottom water implies an inefficient biological pump and low values of sequestered CO_2 in the deep ocean.

We now emphasize the importance of equatorial and subtropical Pacific productivity during “biogenic blooms” rather than high pre-formed nutrient content as a key driver of intermediate and deep water \$\delta^{13}\text{C}\$ variability (see discussion on Lines 246-273).

Lines 253-255: This effect additionally becomes accentuated with an increased carbon sequestration flux, for instance, during periods when monsoonal winds strengthen, as shown in the Site 1146 late Pleistocene record (Suppl. Fig. 12). This would imply that local carbon flux plays an important role in the global storage of carbon in the deep ocean- an idea that is undercut elsewhere in the discussion that emphasizes physical and biochemical aspects of deep water formation as the primary controls on deep ocean carbon sequestration. Likewise, equatorial Pacific productivity generally is neglected in global ocean carbon cycle models which instead emphasize productivity and nutrient utilization in areas of deep and intermediate water formation.

We accept that our previous interpretation was over-simplified. In this revised version of the paper, we modified our original interpretation and revised the text. We now emphasize that the sequestration efficiency of the biological pump in the low latitude Pacific Ocean plays a fundamental role in the global storage of carbon in the deep ocean. Thus, we no longer consider that physical and biochemical aspects of deep water formation are the sole or main controls on deep ocean carbon sequestration (Lines 282-297).

Lines 306-309: “Renewed 100 kyr eccentricity forcing after 7.2 Ma, combined with high-amplitude variations in obliquity, probably drove the successive rebounds between 7.2 and 7.0 Ma, but did not succeed in reversing the cooling trend.” I would avoid using the term “forcing” with “eccentricity” because the insolation variations driven by eccentricity are so small that it is very hard to imagine they directly force the climate. Instead, the dominant view is that nonlinearities in climate feedbacks to precessional variations end up producing an output with the eccentricity modulation imprint.

Text was modified to address this criticism (Lines 311-313).

Lines 330-333: “In particular, our results show that ice volume was not the primary driver of global climate and that low latitude processes including monsoonal winds, large-scale ocean circulation and productivity played a fundamental role in setting late Miocene climate trends. “

While one can agree that the present study uncovers very interesting variability in proxies related to low latitude processes, it hasn't demonstrated that these “set” late Miocene climate trends. In fact, subsequent text proposes the global carbon cycle as the driver.

Text was rephrased (Lines 337-340).

Reviewer #3 (Remarks to the Author):

There is no doubt that the manuscript by Holbourn et al presents high quality geochemical data sets that reflect late Miocene climate evolution in unprecedented detail. As a result, the manuscript contains 23 figures, eight in the main text and 15 in the supplemental information. In order to make such an enormous amount of information readable, results need to be communicated effectively by following a consistent structure. However, this manuscript is difficult to follow at times. Many times I am not sure whether or not I am correctly understanding what the authors are trying to say. So I am unsure whether or not my comments regarding the scientific content are valid and pertinent. My review provides some examples of ambiguities, and I think the authors should revise all of the manuscript with an eye for clarity.

To address these criticisms:

- 1) we substantially re-organized the paper and revised the text, in particular the structure of the Results sections entitled “Temporal trends in benthic and planktic stable isotopes” and” Evolution of mixed layer temperatures” and clarified the text.
- 2) we also re-organized the discussion and modified the text to provide a more logical succession of arguments and to emphasize the main findings from our study.
- 3) we completely re-organized the supplemental information and streamlined the text into 4 Notes that address specific topics supported by relevant supplementary figures.
- 4) we merged several figures (for instance, the Drury et al. data are now included in the main Fig. 6) and we also deleted redundant supplementary figures, thus reducing the total number of main figures from 8 to 7 and the number of supplementary figures from 15 to 10.

Figure call-outs: One problem with the manuscript organization is that the text refers to specific information presented in figures, but then calls out to a range of figures without specifying in which panel the information can be found. Conversely, figures contain panels that are not addressed in the same context as

other panels of the same figure. I feel like the reader is being asked to mentally fill in these gaps, which is time consuming.

An example of confusing paragraph structure:

The topic sentence on line 97 points to differences in long-term trends between benthic and planktonic d18O records, as well as coherence between them at the obliquity-scale. Then the authors simply call out to Figure 2 as well as Supplemental Figures 3-4 (line 98). The reader has to search Figure 2 for the correct panels (B and C), and is also left to search for the specific differences in the long-term trends between these two records. The text does not provide any supporting statements about how the long-term trends shown in Figure 2 are different.

We re-organized and substantially revised this section of the results and now refer to individual figure panels, where appropriate.

- 1) We first present the oxygen isotope results, pointing out differences in their long- and short-term variability (Lines 103-131).
- 2) We then report on the coherence of benthic and planktic d18O and d13C at orbital bands (Lines 132-142) and we additionally provide a summary of coherence and phase relationships in Supplementary Note 2.
- 3) In the final paragraph, we describe benthic and planktic d13C trends and the evolution of the gradient between planktic and benthic d13C (Lines 143-155).

Supplemental Figure 3 presents the results from the spectral analysis. However, it only shows power spectra, the text on the other hand, refers to “high coherence” (line 98). The coherence between benthic and planktic d18O is not shown in this figure (despite their response in the rebuttal to be showing this). Supplemental Figure 4 shows the time series of both records in panels A and B with long-term trends superimposed together with a comparison of the d18O records to corresponding d13C records.

We now provide both Blackman-Tukey and wavelet cross spectral analyses in (Suppl. Figs. 5 and 6) to document coherence and phase relationships between benthic and planktic d18O. We refer to these on Lines 132-142 and in Supplementary Note 2.

In my mind, Supplemental Figure 4 presents the information that fits with the main text of this paragraph (beginning on line 97). Panels A and B very nicely illustrate similarities and differences in the planktic and benthic long-term d18O trends (they are illustrated by a smoothed curve). Then, if the topic sentence of this paragraph is to stand as is (first mentioning the long-term trend and then the short-term trend), it should be followed first by the supporting points regarding the long-term trends second the supporting points regarding the short-term trends. The orbital-scale variations should be addressed calling out to a figure that shows the cross correlation. And, in order to fit the d13C into this paragraph, the topic sentence could be modified to read: “Benthic and planktic stable isotope

records exhibit....”In this manner the pertaining d18O and d13C trends can be fully described before addressing the orbital-scale variations.

Line 107: It is not clear to me what the standard deviations refer to.
Are they standard deviations of averages over discrete time intervals?

Yes, they are. The time windows are now specified in the caption of Suppl. Fig. 4

Line 128: Sentence and call out to Supplemental Figure 7: Is it possible that the color scale of the coherence is wrong? As is, it looks like the background color yellow is also the color that defines highest coherence? It would suggest that the records are highly coherent everywhere except in the well-defined blue areas? The blue, circled areas are those of lowest coherence? The captions does point out, however, that the circled regions (which are blue) are 5% significance against red noise.

We revised the caption of Suppl. Fig. 6 to clarify that black contours, which demarcate 5 % significance level against red noise, circle yellow areas and not blue areas. This is particularly clear in the lower panel of Suppl. Fig. 6, which shows overall low coherence between benthic and planktic d18O. There is no background color. The scale on the right side of Suppl. Fig. 6 indicates coherence: highest coherence is yellow, lowest coherence is blue.

The manuscript text states that at the 41 kyr periodicity, the planktic and benthic d13C values are coherent throughout the time interval. But in the Supplemental Figure 7, the blue regions are far less prominent compared to the yellow background region in the roughly 30-64 kyr band. However, in the precession band, the blue regions dominate. So, is this a typo on line 129?

See response to previous comment.

Lines 179-187: This part of the discussion read more like results. It provides detail that should be described in the context of the description of the time series.
This part of the discussion was transferred to Results.

Lines 269-272: Figure 7 shows the entire benthic d13C and d18O records from Site 1146. How do these records illustrate “re-organization of the SE Asian monsoon”? And, why is there a label “D” in the lower right hand corner?

This misleading sentence was deleted.

Lines 283-284: “in-phase” relationship between benthic d18O maxima and planktic d18O increases: This sentences is confusing. First, this statement implies that cross spectral analyses were conducted, the authors call out to Figure 5, which shows time series.

We replaced “in phase” by “coincide”, to avoid confusion (Line 223). In fact, Blackman-Tukey and wavelet cross spectral analyses (Suppl. Figs. 5 and 6) show that benthic and planktic $\delta^{18}\text{O}$ generally exhibit relatively low coherence

and variable phase relationship at all main orbital bands. Benthic d18O maxima are only in phase with planktic maxima during prominent TG events between 6.1 and 5.5 Ma. These events do not exhibit a simple cyclic pattern.

Second, I am not sure what the authors mean. As written, the sentence implies that planktic d18O values still increase while benthic d18O values have already reached a maximum. In a cross spectral analysis such a relationship would show up as a lead/lag, should it be statistically significant. On the other hand, the same sentence also says that the records indicate synchronous variations between deep water d18O and NH climate. But looking at the planktic and benthic d18O records, the two are not always synchronous. Some of the maxima occur at the same time in both records (TG22 and TG 20), but others are not (TG4). I think the wording needs to be more precise.

We revised the text, including reference to Fig. 5 (Lines 220-225). The inphase relationship benthic and planktic d18O only occurs during TG events within the interval 6.1 to 5.5. Ma, which represents the coldest phase in our record. In fact, increases in benthic and planktic d18O are synchronous during TG 20-22 and TG12-14 (between 6.1 and 5.5 Ma), whereas T8 and TG4 show a lead of benthic d18O (warm phase after 5.5 Ma).

Reviewers' comments:

Reviewer #1 (Remarks to the Author):

The manuscript has improved through the revisions process. The title is also an improvement.

The data remain at high resolution and are of high quality. I am concerned that the secular trend in Mg/Ca seawater issue (etc. I'm thinking of Evans and Muller's work) and subsequent fact that the absolute value of SSTs are incorrect (but relative changes are fine) will be lost on most readers and they will be left thinking that temperatures in the region were cooler than today. I wish there was a better way to handle this issue, perhaps by adopting more recent approaches pioneered by Evans. It's not so important that the exact number is right so much as the overall reference value is closer to what it ought to be (the author's conjecture +1,1.5C).

The interpretation of the data are still without a suitable quantitative physical basis. For example, does a change in CO₂ in Miocene climate models cause the kind of changes they propose? It's irrelevant here whether the models are perfect. If the models predict similar changes... then great! Models match data and the paper is on solid physical foundations. If the models and data disagree, then let the authors say so, perhaps provide 'targets' as they say for improvement.

But, without any discussion of what physical models suggest for this time interval it is unclear whether the CO₂, cooling, winter monsoon argument they provide makes sense. I'll note that they utilize models repeatedly throughout the paper (for example to justify arguments about the interpretation of CO₂ and biological pump; or sea level curves) whenever geochemistry or ice are involved. So it's clear that the authors that data and models must be synthesized and are synergistic, but this does not extend to climate models. Very strange.

This remains a serious weakness in this paper and one that would take one day, one paragraph, and four references to fix.

Reviewer #2 (Remarks to the Author):

I read the comments of Reviewers 1 & 3 closely, as well as the author's responses. The reviewers comments are insightful and generally cohere on the tendency of earlier versions of the manuscript to make vague claims not closely supported by the data presented here. The present version of the text actually seems substantially more revised than the first revision and I think this is a good thing. The authors have clearly heeded Reviewer 1 by more clearly articulating the central claims of the paper and where it provides new information. In particular, the arguments based on analysis of stable isotopic records are substantially deeper and sharper than previous versions of the paper. Additional focus on differences between planktic and benthic records, and an argument for NH enhancement of late Miocene TG coolings/glaciations is justified by the marvelous data sets presented here. Other new arguments take our understanding of late Miocene variability into new, more highly resolved territory: the orbital configuration ~7.2 Ma was similar to 2 previous Miocene cooling episodes. Overall, I feel that the current paper has found a focus that is concordant with an emerging view that late Miocene climatic cooling occurred with little change in ice volume in the Antarctic. The close link here of marine and terrestrial proxies in the Asian monsoonal region represents a further satisfying synthesis. I would recommend publication, with the editor's discretion in asking the authors for minor changes for clarity or style

Reviewer #3 (Remarks to the Author):

This third version of the manuscript is much improved. I can now appreciate the full significance of

the study. The results and discussion better communicate the novel insights gained from these excellent, high resolution records.

There are still some lingering issues, such as vague call-outs to Figures (despite the authors saying in their rebuttal that they fixed this). Also, to my reading, there are a few rather vague statements that don't seem to follow from the context. But overall, the paper is now well organized and much more succinct. Below I provide some more targeted feedback, which I hope will help, but may not be must-do kind of changes.

Line 20: A late Miocene $\delta^{18}O$ increase between ~7-7.5 Ma has been observed and discussed in other studies, see Hodell et al., 2001 (Paleoceanography), Billups, 2002 (PPP).

Line 22: I don't think that this statement is necessarily true. Just because one follows the other in sequence does not necessarily mean that the former is causing the latter.

All in all, I don't think the abstract is summarizing the most salient points of the paper? It seems general.

Line 29: The statement is not necessarily incorrect, one version or another of it is often read in the context of introducing studies about Pliocene climate change. The late Miocene does differ from the modern in a number of tectonic boundary conditions (Central America Seaway, narrowing of the Straits of Gibraltar, opening of the Bering Straits, the Indonesian Throughflow, see for example Brierley and Fedorov (2016, EPSL)). Because these have effects on climate, late Miocene warmth is perhaps not an analogue for future warmth. But that does not mean that this interval of time is not valuable in understanding the relationships between the carbon cycle and climate, just not exactly the way as implied in this sentence?

Line 60-63: what about tectonic changes? Those can affect circulation etc.

Line 65: Missing word 'using' in the parentheses?

Line 67: comma after 'changes'

Line 96: Fig. 2A?

Line 98-100: I think this sentence would be a great topic sentence for the paragraph. I think it would provide good context for this study to highlight that the authors have already worked on this site and are thus familiar with it.

Line 104: which panels of Fig. 2? Panels are provided for the supplementary figures, why not for the ones in the main text?

Line 115: Fig. 2C?

Line 132-136: The wording of the statistical treatment of the results seems odd to me. Coherence is referred to as low, but I think they mean the value for non-zero coherence is low, which means that coherence above it at, e.g., $k=0.9$, is significant. I think it would be more straight forward to just tell the reader whether the records are significantly coherent above the 80% or 95% levels.

Also, it is not clear to me why they use the Blackman Tuckey for the benthic and planktic $\delta^{18}O$ while they use cross wavelet analyses for the respective $\delta^{13}C$ records?

Lines 151, 152, 159: which panels?

Line 167: I do agree that not making a correction and focusing on "relatively short-term changes"

is the best approach. However, the authors should then follow through and keep to describing/discussing relative temperature changes. For example, the text on line 176 should be revised to reflect temperature changes (e.g., cooling by ~2 degrees C). Same thing on line 181 and 182.

Line 170: thermocline temperatures? Should it read mixed layer temperatures (e.g., see line 157, the section title)?

Line 188: This is an example of a vague sentence. How do the d18O differences reflect the Asian landmass specifically? In the order it is written, benthic d18O refers to Asian landmass and planktic d18O refers to the Antarctic ice sheet. More importantly, as written, this seems to be more of a sentence that concludes observations/interpretations, not introduces them. It does not follow from any information given thus far in the text (maybe I missed it). Perhaps this could be fixed by saying that the differences in the trends reflect a decoupling of regional hydrography from global climate/ice-sheet evolution. Then the follow-up paragraphs make the argument that the regional signal is related to the Asian landmass/monsoon?

Line 222: Another example of a sentence that as written does not make sense to me: Benthic and planktic d18O maxima coinciding: how does this reflect deep-water d18O and NH climate? Do the authors mean to say that the Asian monsoon stands for NH climate?

In addition, maybe refer to the isotope stages in the text as those are labeled in the figure? It would help draw the eye to the right place in the figure.

Line 225: which panel? And, Figure 5 only goes to 6.0 Ma, while the text refers to observation to 6.1 Ma.

Line 226: In my mind, the word 'transient' does not go with 'massive'. Maybe just delete 'transient'?

Line 227: Is there evidence for Arctic sea ice or is this conjecture?

Line 259: respective panels?

Line 307: Figure 7B: Would it be possible to simply overlay the eccentricity and obliquity curves onto the d18O record? For example, the left column of the figure would be d18O and eccentricity, the right panels d18O and obliquity? I think it would more effectively demonstrate this point.

We thank the reviewers for critical comments, which considerably helped us to improve the ms.

Please note that line numbers refer to revised text with accepted changes.

Reviewer #1 (Remarks to the Author):

The manuscript has improved through the revisions process. The title is also an improvement.

The data remain at high resolution and are of high quality. I am concerned that the secular trend in Mg/Ca seawater issue (etc. I'm thinking of Evans and Muller's work) and subsequent fact that the absolute value of SSTs are incorrect (but relative changes are fine) will be lost on most readers and they will be left thinking that temperatures in the region were cooler than today. I wish there was a better way to handle this issue, perhaps by adopting more recent approaches pioneered by Evans. It's not so important that the exact number is right so much as the overall reference value is closer to what it ought to be (the author's conjecture +1,1.5C).

To provide more realistic temperature estimates that take into account secular changes in Mg/Ca seawater, we added an additional axis to show both uncorrected and corrected mixed layer temperatures in Figs. 3–5 and Suppl. Fig. 9C.

We also modified the text (Lines 161–176), highlighting differences with present day values and comparing two alternative approaches (Evans et al., 2016 and Lear et al., 2000, 2015) to derive temperature corrections.

The interpretation of the data are still without a suitable quantitative physical basis. For example, does a change in CO₂ in Miocene climate models cause the kind of changes they propose? It's irrelevant here whether the models are perfect. If the models predict similar changes... then great! Models match data and the paper is on solid physical foundations. If the models and data disagree, then let the authors say so, perhaps provide 'targets' as they say for improvement.

But, without any discussion of what physical models suggest for this time interval it is unclear whether the CO₂, cooling, winter monsoon argument they provide makes sense. I'll note that they utilize models repeatedly throughout the paper (for example to justify arguments about the interpretation of CO₂ and biological pump; or sea level curves) whenever geochemistry or ice are involved. So it's clear that the authors that data and models must be synthesized and are synergistic, but this does not extend to climate models. Very strange.

This remains a serious weakness in this paper and one that would take one day, one paragraph, and four references to fix.

We followed this suggestion and added a section on CO₂ forcing in the discussion (Lines 348–380), which integrates the outcomes of several modeling studies of late Miocene (van de Wal et al., 2011; Bradshaw et al., 2012, Stein et al., 2016; Knorr et al., 2011) and Pliocene climates (Fedorov et al., 2013, 2015; Burls and Fedorov, 2014a–b). These simulations emphasize the amplifying role of positive feedbacks in driving climate change, despite relatively small CO₂ changes during the Miocene and Pliocene. These modeling results back up our interpretations that the late Miocene ocean–climate system was highly dynamic and that multiple feedbacks in concert with a

modest decline in pCO₂ (Lines 373–380) drove late Miocene cooling and monsoonal change.

Reviewer #2 (Remarks to the Author):

I read the comments of Reviewers 1 & 3 closely, as well as the author's responses. The reviewers comments are insightful and generally cohere on the tendency of earlier versions of the manuscript to make vague claims not closely supported by the data presented here. The present version of the text actually seems substantially more revised than the first revision and I think this is a good thing. The authors have clearly heeded Reviewer 1 by more clearly articulating the central claims of the paper and where it provides new information. In particular, the arguments based on analysis of stable isotopic records are substantially deeper and sharper than previous versions of the paper. Additional focus on differences between planktic and benthic records, and an argument for NH enhancement of late Miocene TG coolings/glaciations is justified by the marvelous data sets presented here. Other new arguments take our understanding of late Miocene variability into new, more highly resolved territory: the orbital configuration ~7.2 Ma was similar to 2 previous Miocene cooling episodes. Overall, I feel that the current paper has found a focus that is concordant with an emerging view that late Miocene climatic cooling occurred with little change in ice volume in the Antarctic. The close link here of marine and terrestrial proxies in the Asian monsoonal region represents a further satisfying synthesis. I would recommend publication, with the editor's discretion in asking the authors for minor changes for clarity or style

Reviewer #3 (Remarks to the Author):

This third version of the manuscript is much improved. I can now appreciate the full significance of the study. The results and discussion better communicate the novel insights gained from these excellent, high resolution records.

There are still some lingering issues, such a vague call-outs to Figures (despite the authors saying in their rebuttal that they fixed this).
Now fixed.

Also, to my reading, there are a few rather vague statements that don't seem to follow from the context. But overall, the paper is now well organized and much more succinct. Below I provide some more targeted feedback, which I hope will help, but may not be must-do kind of changes.

Line 20: A late Miocene d18O increase between ~7–7.5 Ma has been observed and discussed in other studies, see Hodell et al., 2001 (Paleoceanography), Billups, 2002 (PPP).

We revised the abstract to better highlight new findings and avoid vague statements.

Line 22: I don't think that this statement is necessarily true. Just because one follows the other in sequence does not necessarily mean that the former is causing the latter.

Agreed. We do not imply causality here, we only summarize main findings (Lines 22–23).

All in all, I don't think the abstract is summarizing the most salient points of the paper? It seems general.

Abstract now revised to avoid vague, general statements (Lines 15–30).

Line 29: The statement is not necessarily incorrect, one version or another of it is often read in the context of introducing studies about Pliocene climate change. The late Miocene does differ from the modern in a number of tectonic boundary conditions (Central America Seaway, narrowing of the Straits of Gibraltar, opening of the Bering Straits, the Indonesian Throughflow, see for example Brierley and Fedorov (2016, EPSL)). Because these have effects on climate, late Miocene warmth is perhaps not an analogue for future warmth. But that does not mean that this interval of time is not valuable in understanding the relationships between the carbon cycle and climate, just not exactly the way as implied in this sentence?

We agree and modified the sentence. We no longer state that the late Miocene is a potential analog of future conditions on Earth. Instead, we highlight the fact that this interval provides the opportunity to investigate processes operating during warm climate states (Lines 36–38).

Line 60–63: what about tectonic changes? Those can affect circulation etc.
Now listed (Line 64).

Line 65: Missing word 'using' in the parentheses?
Now added (Line 68).

Line 67: comma after 'changes'
Done.

Line 96: Fig. 2A?
Done. We now specify relevant panel(s), when referring to main figures throughout the text.

Line 98–100: I think this sentence would be a great topic sentence for the paragraph. I think it would provide good context for this study to highlight that the authors have already worked on this site and are thus familiar with it.
We transferred the sentence to the Introduction to provide context for this new study, as suggested by the reviewer (Lines 71–72).

Line 104: which panels of Fig. 2? Panels are provided for the supplementary figures, why not for the ones in the main text?
Done.

Line 115: Fig. 2C?
Done.

Line 132–136: The wording of the statistical treatment of the results seems odd to me. Coherence is referred to as low, but I think they mean the value for non-zero coherence is low, which means that coherence above it at, e.g., $k=0.9$, is significant. I

think it would be more straight forward to just tell the reader whether the records are significantly coherent above the 80% or 95% levels.

We modified the text to clarify this point (Lines 148–158).

Also, it is not clear to me why they use the Blackman Tuckey for the benthic and planktic d18O while they use cross wavelet analyses for the respective d13C records?

We were surprised by this comment, as Blackman Tukey and cross wavelet analyses both for benthic and planktic d18O and for benthic and planktic d13C are presented in Suppl. Figs. 5 and 6. The comment does not seem to apply to these figures.

Lines 151, 152, 159: which panels?

Panels now consistently specified through text.

Line 167: I do agree that not making a correction and focusing on “relatively short-term changes” is the best approach. However, the authors should then follow through and keep to describing/discussing relative temperature changes. For example, the text on line 176 should be revised to reflect temperature changes (e.g., cooling by ~2 degrees C). Same thing on line 181 and 182.

To provide more realistic temperature estimates that take into account secular changes in Mg/Ca seawater, we added an additional axis to show both uncorrected and corrected mixed layer temperatures in Figs. 3–5 and Suppl. Fig. 9C.

We also modified the text (Lines 161–176), highlighting differences with present day values and comparing two alternative approaches (Evans et al., 2016 and Lear et al., 2000, 2015) to derive temperature corrections.

We followed this suggestion, whenever possible. However, we feel it is important to retain the statistical data defining trends in the results section (Lines 176–189). We therefore refer to uncorrected values as well as relative temperature changes in this section.

Line 170: thermocline temperatures? Should it read mixed layer temperatures (e.g., see line 157, the section title)?

Now corrected.

Line 188: This is an example of a vague sentence. How do the d18O differences reflect the Asian landmass specifically? In the order it is written, benthic d18O refers to Asian landmass and planktic d18O refers to the Antarctic ice sheet. More importantly, as written, this seems to be more of a sentence that concludes observations/interpretations, not introduces them. It does not follow from any information given thus far in the text (maybe I missed it). Perhaps this could be fixed by saying that the differences in the trends reflect a decoupling of regional hydrography from global climate/ice-sheet evolution. Then the follow-up paragraphs make the argument that the regional signal is related to the Asian landmass/monsoon? We modified the text as suggested (Lines 198–201).

Line 222: Another example of a sentence that as written does not make sense to me: Benthic and planktic d18O maxima coinciding: how does this reflect deep-water d18O and NH climate? Do the authors mean to say that the Asian monsoon stands for NH climate?

We modified the text to clarify this point (Lines 232–237).

In addition, maybe refer to the isotope stages in the text as those are labeled in the figure? It would help draw the eye to the right place in the figure.

Done.

Line 225: which panel? And, Figure 5 only goes to 6.0 Ma, while the text refers to observation to 6.1 Ma.

Panels now consistently specified through text. We changed 6.1 to 6.0 Ma in the text to match the figure cited.

Line 226: In my mind, the word 'transient' does not go with 'massive'. Maybe just delete 'transient'?

Done.

Line 227: Is there evidence for Arctic sea ice or is this conjecture?

Yes, we refer to published evidence on Lines 237–239.

Line 259: respective panels?

Panels now consistently specified through text.

Line 307: Figure 7B: Would it be possible to simply overlay the eccentricity and obliquity curves onto the d18O record? For example, the left column of the figure would be d18O and eccentricity, the right panels d18O and obliquity? I think it would more effectively demonstrate this point.

Done. Thank you for this great suggestion. It does clarify the figure!

REVIEWERS' COMMENTS:

Reviewer #1 (Remarks to the Author):

The paper looks good to me now.

REVIEWERS' COMMENTS:

Reviewer #1 (Remarks to the Author):

The paper looks good to me now.

We thank all reviewers for helpful, constructive comments.